microbiology, ecology

gut microbiome, colonization resistance, competition, community interactions

**Author for correspondence:**
Michael Baumgartner
e-mail: michael.baumgartner@env.ethz.ch

# Microbial community composition interacts with local abiotic conditions to drive colonization resistance in human gut microbiome samples

Michael Baumgartner, Katia R. Pfrunder-Cardozo and Alex R. Hall

Institute of Integrative Biology, Department of Environmental Systems Science, ETH Zürich, 8092 Zürich, Switzerland

MB, 0000-0002-4158-460X; ARH, 0000-0002-3654-1373

Biological invasions can alter ecosystem stability and function, and predicting what happens when a new species or strain arrives remains a major challenge in ecology. In the mammalian gastrointestinal tract, susceptibility of the resident microbial community to invasion by pathogens has important implications for host health. However, at the community level, it is unclear whether susceptibility to invasion depends mostly on resident community composition (which microbes are present), or also on local abiotic conditions (such as nutrient status). Here, we used a gut microcosm system to disentangle some of the drivers of susceptibility to invasion in microbial communities sampled from humans. We found resident microbial communities inhibited an invading *Escherichia coli* strain, compared to community-free control treatments, sometimes excluding the invader completely (colonization resistance). These effects were stronger at later time points, when we also detected altered community composition and nutrient availability. By separating these two components (microbial community and abiotic environment), we found taxonomic composition played a crucial role in suppressing invasion, but this depended critically on local abiotic conditions (adapted communities were more suppressive in nutrient-depleted conditions). This helps predict when resident communities will be most susceptible to invasion, with implications for optimizing treatments based on microbiota management.

## 1. Introduction

Biological invasions have major impacts on ecosystem function and diversity [1]. Many factors can influence whether invading species successfully colonize new communities [2,3], making it a key challenge in ecology to understand what drives the outcome of invasions. One ecosystem where susceptibility to invasion has direct impacts on human health is the intestinal microbiome. Here, the resident microbial community protects against infection by pathogens, and disturbance can lead to opportunistic invasion [4–7]. What determines the ability of resident gastrointestinal communities to suppress colonization by invading species (colonization resistance [8,9])? This question is central to our basic understanding of how microbial communities respond to invasion, for explaining variable susceptibility to infectious disease, and predicting the success of microbiota-based therapies such as faecal microbial transplantation [10]. Some physiological mechanisms by which individual resident taxa impact colonization resistance have been characterized [11], including direct interactions among microbes and indirectly via interplay with the host immune system [12]. Other recent studies have demonstrated the net effect of entire microbial communities on colonization success of invading strains [13,14]. However, it is

unclear whether such effects depend primarily on which resident organisms are present (e.g. particular strains or species), in which case they could potentially be predicted from metagenomic data, or also on local abiotic conditions, such as nutrient availability. Disentangling these factors is challenging, because in natural microbiomes they are intertwined: changing community structure modifies the local micro-environment, and vice versa.

Depending on which types of microbial interactions are most important, we can expect susceptibility of resident microbiota to invasion to depend on community composition and/or local abiotic conditions in various ways. For example, if invading strains are inhibited by toxins produced by a subset of the resident bacteria, colonization resistance will rely primarily on the presence of particular taxa (those encoding invader-inhibiting mechanisms). There is support for such mechanisms in that the type VI secretion system, encoding contact-dependent growth inhibition of other strains, is widespread in the common gut phylum Bacteroidetes [15–17]. Similarly, the production of narrow-spectrum antibacterial toxins is common in the gut microbiome, and commensals producing bacteriocins, such as *Escherichia coli* and *Bifidobacterium* spp., can suppress pathogens compared to non-producing mutant strains [18,19]. By contrast, other mechanisms of colonization resistance should be less contingent on community composition. For example, resource competition between resident microbiota and invading strains does not necessarily require a specific set of taxa to be present, only that the resident community is diverse and dense enough to scavenge resources shared with the invader. In support of a role for resource competition, after disturbance of microbiota by antibiotic treatment the availability of free sugars and amino acids can increase, and subsequently be exploited by opportunistic pathogens like *Salmonella enterica* serovar Typhimurium or *Clostridium difficile* [20,21]. Crucially, we can expect this type of colonization resistance to depend strongly on local abiotic conditions, in that resource competition will be most effective at inhibiting invading strains when shared resources become scarce [22]. Other types of resource competition, such as that for resources produced as metabolic by-products of particular resident microbes [23] could nevertheless be sensitive to community composition as well.

In nature, colonization resistance is probably influenced by multiple mechanisms simultaneously. For example, *Bifidobacterium* spp. acidifies the local environment and antagonize some opportunistic pathogens, but also compete for resources with other members of the community [12]. Furthermore, while colonization success is ultimately determined by the net population growth of the invading strain, component replication and death rates may be affected differently depending on the relative contributions of different mechanisms (e.g. resource competition may inhibit replication, whereas direct killing will increase death rates). Despite this complex picture of potentially overlapping, diverse mechanisms, the net effect of resident communities on population growth of invading strains does appear to be sensitive to individual community-level properties, such as microbiota composition in mouse models [24]. This is promising because it indicates such properties could potentially be manipulated in ways that improve colonization resistance, even if the various mechanistic drivers are challenging to isolate. Therefore, as a first step toward understanding how community composition and abiotic conditions interact to influence colonization resistance, measuring their effects on

net colonization success in controlled conditions would improve our ability to manipulate these aspects in treatment and to investigate the component mechanisms involved.

We aimed to test whether colonization resistance in individual human microbiome samples is affected by changes over time in microbiota composition, local abiotic conditions (in particular, nutrient limitation), or both. We did this by observing the invasion of human-associated microbiota by a non-resident focal strain in replicated microcosm experiments. We used a gut microcosm system [13] to co-cultivate the invading focal strain with microbiota sampled from three healthy human donors. We used *E. coli* as the focal strain because it is a common gut commensal [25], but also an opportunistic pathogen [26]. By using microcosms with sterilized and 'live' versions of the resident microbiota, we quantified the effect of resident microbiota on population growth of our focal strain. We maintained each microcosm with periodic sampling for 72 h. This allowed us to monitor the effect of resident microbiota both early on and during later stages when we expected the environment to have become relatively nutrient-poor. We also tracked changes in community composition over time, using amplicon sequencing. Finally, by separating resident microorganisms from the supernatant (liquid phase) of individual microcosms and then using these two phases in further experiments, we were able to disentangle the effects of the resident microbiota and the local abiotic conditions on the population growth of the focal strain.

## 2. Material and methods

### (a) Focal bacterial strain (invader)

We used *E. coli* K-12 MG1655 with a streptomycin-resistance mutation (*rpsL* K43R) as our focal strain. Prior to the main microcosm experiment, we inoculated the focal strain for 24 h in anaerobic basal medium [27,28] with some modifications (2 g l$^{-1}$ peptone, 2 g l$^{-1}$ tryptone, 2 g l$^{-1}$ yeast extract, 0.1 g l$^{-1}$ NaCl, 0.04 g K$_2$HPO$_4$, 0.04 g l$^{-1}$ KH$_2$PO$_4$, 0.01 g l$^{-1}$ MgSO$_4$ × 7H$_2$O, 0.01 g l$^{-1}$ CaCl × 6H$_2$O, 2 g l$^{-1}$ NaHCO$_3$, 2 ml Tween 80, 0.005 g l$^{-1}$ haemin, 0.5 g l$^{-1}$ L-cysteine, 0.5 g l$^{-1}$ bile salts, 2 g l$^{-1}$ starch, 1.5 g l$^{-1}$ casein, 0.001 g l$^{-1}$ resazurin, pH adjusted to 7, the addition of 0.001 g l$^{-1}$ menadione after autoclaving) under a constant stream of nitrogen gas, sealed the tubes and incubated at 37°C and 220 r.p.m. in a shaking incubator.

### (b) Faecal samples

The following protocol for obtaining human faecal samples was approved by the ETH Zürich Ethics Commission (EK 2016-N-55). We collected samples on 8 January 2018 from three anonymous, consenting donors and kept them anaerobic approximately 1 h before processing. We re-suspended 10 g of each sample in 100 ml anaerobic peptone wash (1 g l$^{-1}$ peptone, 0.5 g l$^{-1}$ L-cysteine, 0.5 g l$^{-1}$ bile salts, 0.001 g l$^{-1}$ resazurin), stirring for 5 min followed by 15 min of resting to sediment. Fifty millilitres of each of these 10% (w/v) faecal slurries were transferred to 100 ml flasks and autoclaved to prepare sterile slurry. The other 50 ml, with 'live' slurry, were stored at room temperature until further processing, similar to the procedure described in [13].

### (c) Anaerobic batch culture system, sampling and bacterial enumeration

Microcosms consisted of Hungate tubes, individually sealed anaerobic test tubes, as in [13]. We filled each tube with 7.2 ml

basal medium, flushed the head space with nitrogen gas, then autoclaved. We then added 850 µl sterilized slurry (community-free treatments) or 350 µl live slurry and 500 µl sterilized slurry (community treatments; electronic supplementary material, figure S1A). We added the focal strain to each microcosm by adding 8 µl overnight culture (approximately $10^6$ colony-forming units, CFU). For the control treatment (basal medium without faecal slurry), we inoculated the focal strain in basal medium supplemented with 850 µl peptone wash. We incubated all microcosms at 37°C static and took samples after 2, 24, 48 and 72 h. To estimate focal strain abundance, we diluted samples in phosphate-buffered solution (PBS) and plated on Chromatic MH agar (Liofilchem, Roseto, Italy) supplemented with streptomycin (100 µg ml$^{-1}$), before counting CFUs. We initially screened the faecal slurry of each human donor to verify the specificity of our selective plates; this revealed no resident bacteria able to grow on these plates. To estimate total bacterial abundance (including the resident microbiota), we used flow cytometry. We diluted each microcosm sample with PBS and stained it with Sybr green (Life Technologies, Zug, Switzerland), with a final concentration of $10^{-4}$ of the commercial stock solution. We used a Novocyte 2000R (ACEA Biosciences Inc., San Diego, USA) equipped with a laser emitting at 488 nm and the standard filter set-up. Detection of bacteria was based on their signature in a plot of forward scatter versus green fluorescence. This approach detects viable cells with negligible background signal from sterilized slurry, although we note such methods have other limitations, such as possible undercounting of cells in aggregates.

### (d) Supernatant experiment

We extracted the supernatant of each microcosm at the end of the experiment. As a control treatment, we supplemented fresh basal medium with thawed, sterilized slurry from each human donor of the batch culture experiment, the same way as for microcosms at the start of the main experiment in community-free treatments. To extract supernatants from each treatment (community, community-free and control), we transferred 1.5 ml of each culture to a 2 ml tube and centrifuged (10 000 r.p.m., 5 min), before syringe-filtering (0.22 µm pore size). To prepare inocula, we made independent cultures of our focal strain in 27 Hungate tubes each filled with 5 ml anaerobic basal medium and incubated for 24 h at 37°C without shaking. We then started the experiment in an anaerobic chamber by transferring 800 µl of each supernatant to a Hungate tube and inoculating each tube with 5 µl of one of the focal strain cultures, before incubating 24 h at 37°C static. We took samples for CFU counts at the beginning and end, diluting in PBS where necessary before plating on LB agar.

### (e) Colonization resistance of conditioned and fresh communities

To test whether resident microbial communities that were conditioned to the microcosm environment (had been incubated for 72 h; hereafter referred to as 'conditioned' communities) were more resistant than fresh communities against invasion by our focal strain, and whether this effect was contingent on changes in the abiotic conditions over time, we produced various combinations of communities and supernatants (electronic supplementary material, figure S1B and Methods). In summary, we first produced and then froze community samples that were either conditioned (had been incubated for 72 h in the absence of the focal strain, but in the same conditions as in the main microcosm experiment above) or fresh (prepared as at the start of the community treatments above, but without the focal strain). Using frozen community samples here allowed us to directly compare samples from before (fresh samples) and after 72 h cultivation in microcosms (conditioned). There is a risk

some taxa are affected by freeze–thawing, although past work indicates such effects are minimal [29] and frozen slurry contains abundant, species-rich communities [14,30], supported further by our results below. We made and froze three replicate fresh and conditioned community samples per human donor (electronic supplementary material, figure S1). We then thawed these samples, measured/adjusted their total bacterial densities, separated the community in each sample from the liquid phase (supernatant) by centrifugation/filtration, and created all possible combinations of fresh/conditioned community and fresh/spent supernatant for each human donor, in triplicate (electronic supplementary material, figure S1). We then inoculated each community/supernatant combination with the invading focal strain, and measured its population growth over 24 h by plating as described above (further details in electronic supplementary material, Methods).

### (f) Amplicon sequencing

We extracted the DNA for amplicon sequencing with the Power-Lyzer PowerSoil Kit (Qiagen) with some modifications to the manufacturer's protocol. In brief, we thawed samples from 0 h and 72 h from each microcosm of the community treatments of the main experiment and homogenized them by vortexing for 5 min. We transferred 1.5 ml of each sample into a Power Bead Tube and centrifuged at 13 000 r.p.m. for 10 min. We removed the supernatant and repeated this step to concentrate the samples. We then extracted the DNA from these concentrated samples following the manufacturer's protocol. DNA quality and yield was checked with Nanodrop and Qubit. For library preparation and sequencing, we followed the Illumina 16S Metagenomic Sequencing Library preparation guide for the MiSeq Illumina sequencing platform (see electronic supplementary material, Methods). We then used Trimmomatic to filter raw sequencing reads and remove adaptors. We used scripts from Usearch [31] to merge amplicons into pairs, trim primer sites and cluster operational taxonomic units (OTUs). We assigned taxonomy of OTUs using Syntax and the SILVA 16S rRNA database (accessed April 2020). We then used the phyloseq [32] package to visualize and calculate the relative abundances of taxonomic groups on the family level and alpha diversity based on total OTU abundance on the genus level.

### (g) Statistics

To analyse variation of focal strain abundance in the main microcosm experiment, we used a linear mixed effects model (lmer function in R v. 3.5.1 [33]). We excluded the basal medium treatment from the analysis and used time, donor (three levels: human donor 1, 2, 3, indicating the origin of the faecal sample in each microcosm) and community (two levels: with, without, indicating live/sterilized slurry) as fixed factors and replicate as a random effect, with Box Cox transformation of the response variable ($\lambda = 0.173439$) to account for heteroscedasticity. We obtained $p$ values for interaction terms using type II Wald $\chi^2$-tests. For principal coordinate analysis (PCoA) of the amplicon data, we used the ordinate function of the phyloseq package with MDS as the ordination method and Bray–Curtis dissimilarity to create the distance matrix. To test whether supernatants from different treatments varied in their effects on growth of the focal strain, we used analysis of variance (lm function) with focal strain change in CFU per millilitre over time as the response variable, and donor and treatment as factors. We used a linear mixed effects model (lmer function) for testing the effects of community age (fresh versus conditioned) and supernatant age (fresh versus spent), with focal strain growth (total change in CFU count over time) as the response variable, which we log transformed and donor, community age and supernatant age as fixed factors. To account for dependencies between

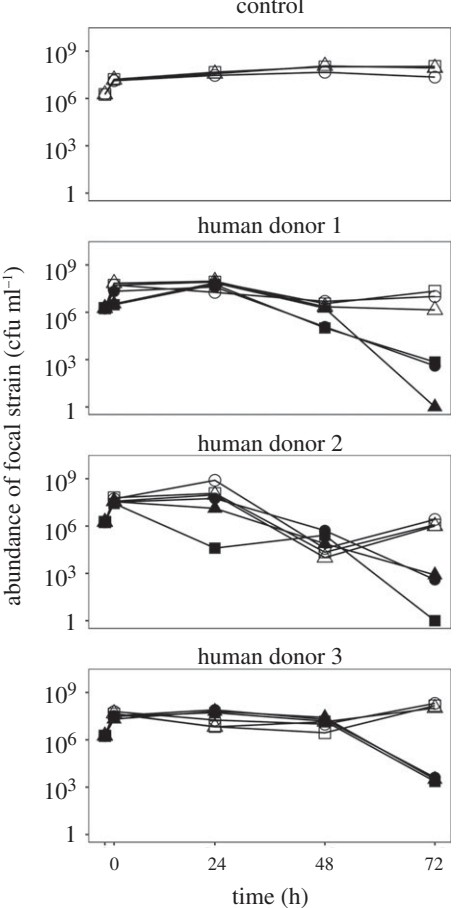

**Figure 1.** Suppressed growth of an invading focal strain in the presence of resident, human-associated microbiota. Each panel shows abundance of the focal *E. coli* strain over 72 h in basal medium only (control; top panel) or in the presence of faecal slurry prepared with samples from one of three healthy human donors (donors 1–3; lower three panels). For each human donor, three replicate microcosms (different symbols) are shown with sterilized faecal slurry (without community; empty symbols) and live slurry including the resident microbiota (with community; filled symbols). Treatments where the focal strain was below the detection limit are shown at 1.

the treatments due to the swapping of supernatant and community samples (each sample was used in two assay microcosms), we assigned a unique identifier (ID) for each community and supernatant sample, and used these IDs as separate random effects. We reduced the model by removing non-significant interactions using *F*-tests.

## 3. Results

### (a) Resident microbiota suppress invasion by a focal *Escherichia coli* strain

We cultivated our focal *E. coli* strain in sterilized and 'live' versions of faecal slurry from three different human donors, measuring focal strain abundance over 72 h with periodic sampling, but without serial passage. We found the presence of resident microbial communities (in live faecal slurries) suppressed focal strain abundance compared to community-free (sterilized) versions of the same faecal slurries (effect of community in linear mixed effects model: $\chi^2 = 41.23$, d.f. = 1, $p < 0.001$; figure 1). This suppressive effect of resident communities was strongest toward the end of the

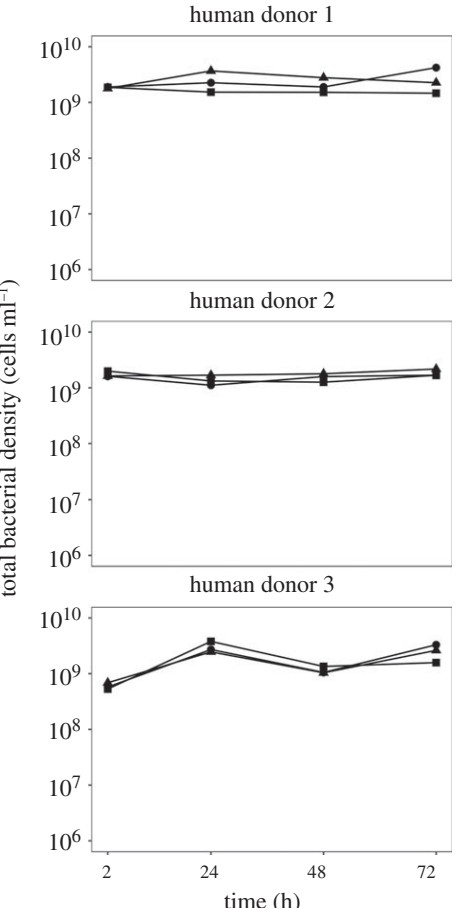

**Figure 2.** High total bacterial abundance over time. Each panel shows total bacterial abundance estimated by flow cytometry for microcosms from the community (live slurry) treatment for one of the three human donors. Replicate microcosms have different symbols.

experiment (community×time interaction: $\chi^2 = 55.24$, d.f. = 3, $p < 0.0001$), with the sharpest decline between 48 h and 72 h, in contrast to the sterilized slurries and control treatment, where focal strain abundance was relatively stable (figure 1). In two community-treatment microcosms, each from a different human donor, suppression was strong enough to push the invading focal strain below our detection limit, amounting to full colonization resistance. Although average focal strain abundance varied among the different donor treatments (effect of human donor in linear mixed effect model: $\chi^2 = 17.27$, d.f. = 2, $p < 0.0001$), community suppression was on average consistent across human donors (community×donor interaction: $\chi^2 = 1.64$, d.f. = 2, $p = 0.44$). Note the decline of focal strain abundance over time was not associated with a general decline of total bacterial population densities (including the resident microbiota): total bacterial density estimated by flow cytometry was stable at high numbers throughout the experiment (figure 2). This shows interactions with resident microbial communities resulted in a decrease in focal strain density over time.

### (b) Resident microbial diversity was maintained over time, but with shifts in relative abundance

The stronger suppression of the invading focal strain we observed at later time points could potentially be explained by changes in the taxonomic composition of resident

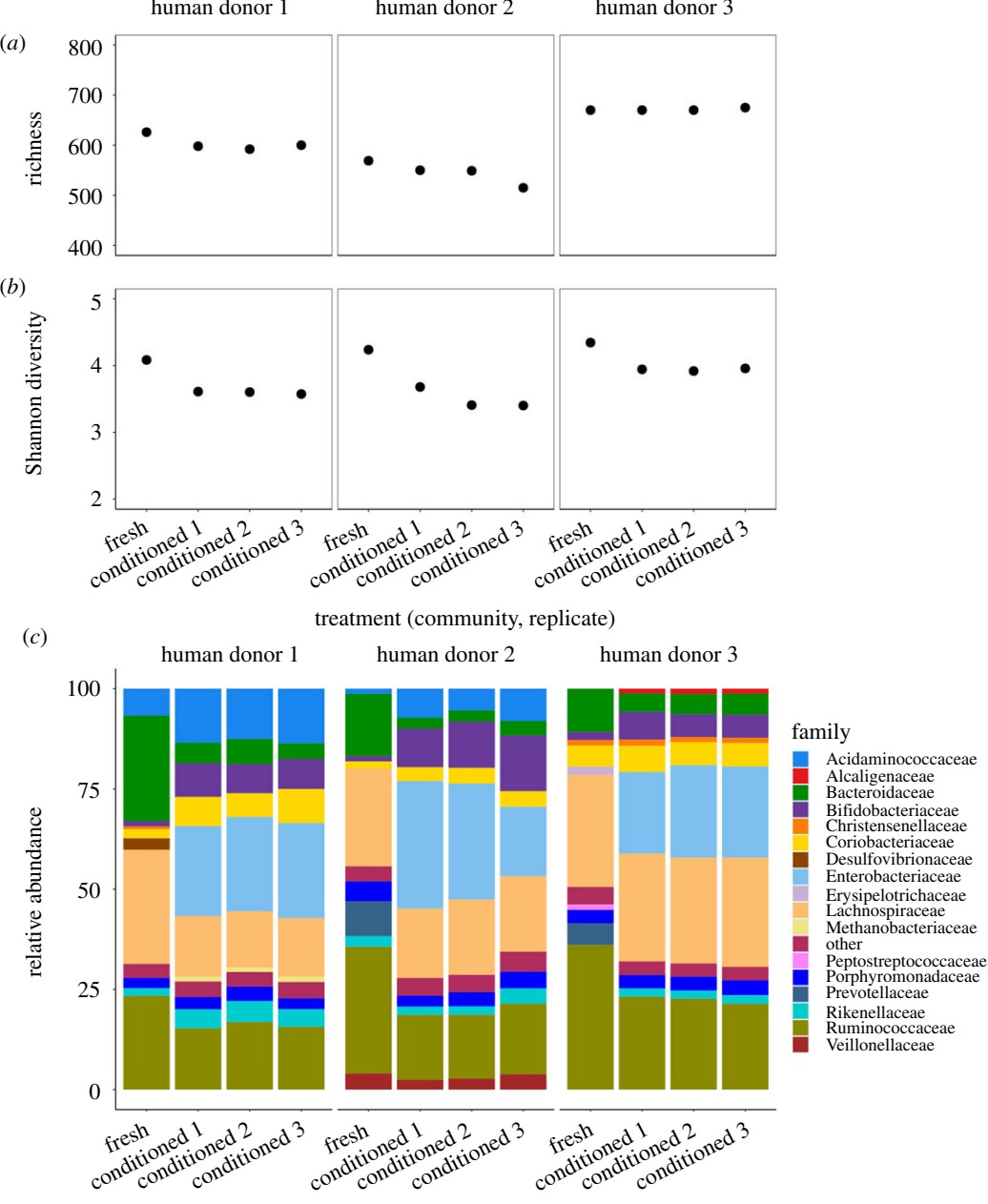

**Figure 3.** High within-sample taxonomic diversity and shifts of relative abundances over time. Taxonomic diversity based on amplicon sequencing of the 16S rRNA gene was estimated by (*a*) richness measured as total number of OTUs on the genus level and (*b*) Shannon's diversity index at the beginning and end of the microcosm experiment. (*c*) Relative abundances of the top 15 families in each microcosm. For each human donor (left, middle, right panels), each of these measures is shown for the fresh faecal slurry (start of the microcosm experiment) and three replicates conditioned microcosms from the community treatments at the end of the microcosm experiment (*x*-axis). (Online version in colour.)

microbial communities. As a first step to investigate this, we tested for changes in taxonomic composition using amplicon sequencing. Taxonomic richness, taken as the number of OTUs at the genus level, of all communities was approximately stable over time (figure 3*a*). However, a measure of taxonomic diversity that accounts for evenness across different groups (Shannon index) showed a decline after 72 h, and this was true for resident microbial communities from all three human donors (figure 3*b*). These shifts in relative abundance were also evident when we looked at the identities of the most abundant taxa. Initially, communities were dominated by Bacteroidaceae, Ruminococcaceae and Lachnospiraceae. Over time, these became less abundant relative to Enterobacteriaceae and Bifidobacteriaceae (figure 3*c*). Despite these changes in relative abundance, the top 10–15 families

were the same after 72 h. A closer look at the taxonomic assignments of the sequencing reads in the Enterobacteriaceae family revealed that almost 100% were assigned as *E. coli* and less than 0.1% were not annotated to the genus level. Note reads assigned to Enterobacteriaceae include those from both resident *E. coli* and the invading focal strain. To gain a rough indication of the abundance of focal strain relative to other *E. coli*, we combined information about the fraction of the total community made up by Enterobacteriaceae (from amplicon sequencing), total community abundance (from flow cytometry), and focal strain abundance (from selective plating). This suggested that in all samples the *E. coli* population was dominated by resident strains and our focal strain contributed less than 0.001% to the total *E. coli* abundance after 72 h (electronic supplementary material, table S1).

PCoA based on Bray–Curtis dissimilarities confirmed that resident microbial communities in different microcosms changed over time in similar ways (along the same axis in electronic supplementary material, figure S2A). This also revealed that communities from human donors 1 and 2 were more similar to each other than to those from human donor 3 (electronic supplementary material, figure S2A). An alternative PCoA analysis excluding Enterobacteriaceae from the dataset and recalculating relative abundances showed a similar qualitative trend in terms of changes over time, suggesting the expansion of this family alone does not explain the shift in community composition (electronic supplementary material, figure S2B). In summary, our analysis of total microbial diversity revealed shifts in relative abundance for certain taxa, such as an expansion of resident *E. coli* strains, and these coincided with increased suppression of the focal strain toward the end of the experiment observed above.

## (c) Changing abiotic conditions alone do not explain focal strain suppression

Another possible mechanism for the observed suppression of the invading focal strain is that the resident microbiota changes the local abiotic environment over time in a way that permits less population growth of the focal strain in the community treatments than in the community-free treatments (e.g. if the resident microbiota causes resource depletion or accumulation of compounds toxic to the focal strain). To test this, we inoculated our focal strain into supernatant extracted by centrifugation and filtration of cultures from the community and community-free treatments at the end of the main microcosm experiment above. Additionally, we inoculated the focal strain into freshly prepared basal medium supplemented with sterilized slurry (unspent medium with slurry thawed from frozen, equivalent to the medium used in community-free treatments at the start of the main experiment) as a control. Our focal strain grew in the supernatants from both the community and community-free treatments (positive net change in abundance over 24 h; figure 4). This suggests in neither treatment had the abiotic environment become toxic to the focal strain. Furthermore, supernatants from both treatments supported a similar amount of population growth of the focal strain (effect of treatment in a model excluding the control treatment: $F_{1,12} = 0.54$, $p = 0.48$). This shows mixed communities comprising the resident community plus the focal strain did not deplete nutrients that support focal strain growth any more than the focal strain did when growing alone. Despite this, the supernatants from both these treatments supported significantly less growth than freshly prepared, sterilized-faecal-slurry medium (effect of treatment, including the control treatment: $F_{2,18} = 946,68$, $p < 0001$). This is consistent with microbial activity in our main microcosm experiment depleting resources that the focal strain uses for growth. We refer to this hereafter as nutrient depletion (supporting less focal strain population growth), although this does not necessarily mean the environment became nutrient-poor for all resident bacteria (not the focal strain). Moreover, this does not explain why we observed suppression of the focal strain in the community treatments compared to the community-free treatments.

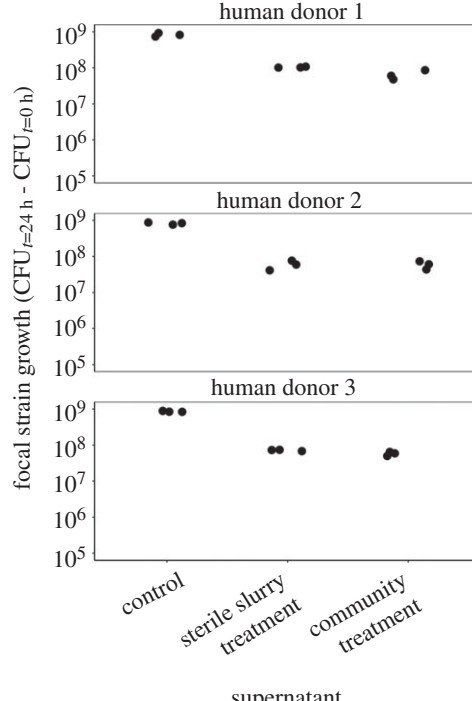

**Figure 4.** Nutrient depletion in both community and community-free treatments. Focal strain growth (change in CFU density over time) in freshly prepared supernatant (control, equivalent to community-free treatment at the start of the main experiment, figure 1), or supernatant from the community and community-free treatments at the end of the main experiment. The three replicates in each treatment had supernatant from independent microcosms of the main experiment.

## (d) Suppressing effects of resident communities depend on local abiotic conditions

Having observed changes in both community composition and the local abiotic environment (nutrient depletion) above, we hypothesized suppression of the invading focal strain toward the end of the main experiment resulted from an interaction between these two factors. That is, we asked whether suppression required communities that were conditioned to our microcosm environment, depleted nutrient status in the microcosm, or both. To do this, we compared fresh and conditioned versions of the same resident microbial communities used above (prepared from frozen faecal slurry, sampled before and after 72 h incubation in the same conditions as in the main experiment, but without the focal strain; schematically illustrated in electronic supplementary material, figure S1) and fresh and spent versions of the local abiotic environment (by extracting supernatant from the same microcosms used to prepare fresh and conditioned communities; electronic supplementary material, figure S1). We then made a fully factorial experiment testing the effects of fresh/conditioned communities and fresh/spent supernatant on focal strain growth (electronic supplementary material, figure S1). This showed focal strain growth was lower in the presence of conditioned communities on average, but only in spent medium (community×medium interaction: $\chi^2 = 179.57$, d.f. = 1, $p < 0.001$; figure 5). This trend was consistent for all three donors, although weaker with the human donor 2 community. This is in line with the main microcosm experiment, where the donor 2 community showed stronger suppression of the focal strain already after 24 h compared

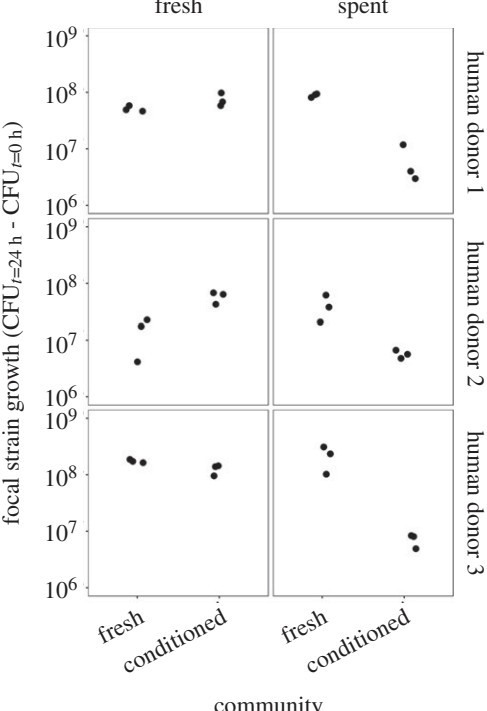

**Figure 5.** Focal strain suppression depends on community composition and the state of the environment. Each panel shows growth over 24 h of the focal *E. coli* strain (in CFU ml$^{-1}$) co-cultivated with a conditioned community (from 72 h-old microcosms; see Methods) or a fresh community (coming directly from faecal slurry) from each human donor. Each co-culture was inoculated into fresh (left column) or spent (right column; supernatant from 72 h-old microcosms) medium. Each point shows an independent replicate.

with communities from donors 1 and 3. In summary, changes in community composition over time made conditioned microbiota more suppressive than fresh microbiota, but this was only observed after the abiotic conditions had become relatively nutrient-poor for the focal strain.

## 4. Discussion

We showed colonization success of an invading lineage depended on an interaction between the taxonomic composition of the resident community and the nutrient status of the local environment. We demonstrated this by monitoring invasion success (in terms of net change in population density) of a strain of the common gastrointestinal species *E. coli* in human-associated microbiome samples. We observed time-dependent suppression of the invading *E. coli*, in some cases amounting to full colonization resistance. This suppression coincided with a change in microbial community composition and declining nutrient status (for the invading focal strain) in the microcosms. However, by splitting the microcosm system into the liquid phase and the resident microbes (by centrifugation and filtration), we showed these two factors interact with each other. Microbial communities that were conditioned to their local microcosm environments were most resistant to invasion, particularly when available resources were scarce, amplifying competition with invading strains. This provides new insights into what makes some communities more susceptible to invasion than others.

The first key implication of our results is the dependency of colonization resistance on both aspects, community

composition and local abiotic conditions. That changing taxonomic composition was linked to altered susceptibility to invasion is promising in terms of predicting colonization resistance of microbiota from individual people. However, translating taxonomic information such as 16S data into predictions about community-level susceptibility to invasion remains a significant challenge [34]. Summary metrics, such as species diversity as we analysed above, may help, and have been correlated with microbiota-related properties such as risk of obesity [35], diabetes [36] and recurrent *Clostridium difficile* infection [37]. However, in our experiment diversity by some measures (Shannon index) declined as communities became more robust to invasion, and by others (richness) remained stable. This is in apparent contrast with the general principle that more diverse ecosystems should be harder to invade [2,38]. This indicates adjustment of communities to local conditions can play a key role in resistance to invasion, even if it is associated with a drop in total diversity by some measures, consistent with there being functional redundancy for some properties [39]. Such adjustment of communities can result from species sorting (here indicated by amplicon data), although we do not rule out an additional role for evolution within individual taxa. Thus, it is probably not simply the case that higher diversity means more colonization resistance. In support, previous studies in mouse models identified consortia of four and 15 commensal species that cleared the intestine from an antibiotic-resistant *Enterococcus faecium* strain and established colonization resistance against *Salmonella enterica* serovar Typhimurium [40,41]. More importantly, our finding that suppressive effects of resident communities were strongly modified by local conditions (different supernatants) suggests sequence data alone, or other information about community composition, are insufficient to accurately predict colonization resistance and information about local ecological interactions is required. This is consistent with past work showing nutrient supplementation can interfere with competitive interactions among resident and invading microbes in mouse microbiota [42,43]. Our results go beyond this to show nutrient status also plays a key role in modulating colonization resistance in human-associated microbiota.

The second key implication of our work is for interventions aimed at improving colonization resistance of individual hosts/patients, such as faecal microbial transplantation [10]. The interaction between community composition and local abiotic conditions we observed indicates colonization resistance resulting from such interventions will depend not only on the type of community that is implanted, but on factors that influence the local micro-environment, such as host diet or physiological status [44]. We found this interaction varied among donors (with donor 2 community samples being relatively suppressive even with fresh supernatant), indicating some person-to-person variation of the relative importance of different drivers of colonization resistance. However, that suppression of the invading focal strain was consistent across donors is also encouraging, indicating some key properties are repeatable across randomly selected healthy-donor communities. This, and our amplicon data, are consistent with the notion of a core microbiome conferring similar functions across healthy individuals, despite variation of the individual taxa present [45].

Part of the taxonomic shift over time in our main experiment was driven by the expansion of Enterobacteriaceae,

the same family our focal strain belongs to. Intraspecific competition between resident and focal *E. coli* may explain at least some of the suppressive effects of resident communities. Consistent with this, we found previously that resident *E. coli* strains sampled from other human donors had a competitive advantage over this focal strain *in vitro* [13]. Turnover of resident and transient *E. coli* clones has recently been observed in samples from humans [46], suggesting such intraspecific competition is also possible in nature. Such strains can carry anti-competitor mechanisms, such as type VI secretion systems [47]. Nevertheless, other taxa also increased in relative abundance, such as Bifidobacteriaceae (figure 3), so we do not exclude there also being a role for interspecific competition. More importantly, the ecological mechanism behind our key results (stronger suppression of an invading strain when the resources it uses for population growth are scarce and the resident community is conditioned to local conditions), is likely not limited to particular strains or species. We note that while our data show a clear community-level effect on net population growth of the invading focal strain, they do not separate community effects on component birth (replication) and death rates. While this is not essential for measuring the variability of invasion success, it would shed light on the types of mechanisms driving the suppression of invaders (killing versus growth inhibition). Possible avenues to separate these processes in future work would be to use plasmid-segregation [48] or mixed-tagging protocols [49] to measure replication and death in human-associated communities.

Our microcosm approach allowed us to observe the invasion of human-associated communities directly, but also imposes limitations. First, we account only for drivers of colonization resistance involving microbial interactions, not those involving the host immune system [12]. Despite this, our findings are consistent with evidence that competition in nutrient-depleted conditions matters for colonization resistance in mice [50–52], although not always [53]. A second limitation is that our centrifuged and filtered supernatants, although clearly containing far fewer bacteria than the pelleted part of each sample, could potentially have contained other bioactive material such as bacteriophages. While we do not exclude the possibility bacteriophages in supernatants could have infected the focal strain in subsequent assays, this seems unlikely to explain our results. Such bacteriophages would not have been present in supernatants from sterile-slurry treatments (figure 4), where we observed similar suppression compared to community-treatment supernatants. Additionally, we previously screened samples collected using a similar design for plaque-forming units with this focal strain, and found none [13]. Finally, it is possible some toxins or inhibitory matter remained attached to the matrix that was pelleted/filtered in preparing supernatants, contributing to the weaker inhibition we observed compared to in live slurry, although this is unlikely to explain the drop in population growth we observed in spent versus fresh supernatant.

In conclusion, our results suggest the outcome of invasion by a new strain or species depends on both the taxonomic composition of the resident human gut microbiota and local abiotic conditions. Resident communities that were conditioned to the microcosm environment were only strongly suppressive in nutrient-depleted conditions. A key challenge for putting such insights into practice in the context of microbiota-based treatments is to identify scenarios (types of infections, health conditions or other biomarkers) where susceptibility to infection can be predicted more strongly from taxonomic information, and interventions that can change the within-host environment in ways that maximize colonization resistance against pathogens.

**Ethics.** Faecal samples were donated by anonymous consenting volunteers and the sampling protocol was approved by the ETHZ Ethics Commission (EK 2016-N-55).

**Data accessibility.** 16S rRNA sequences are deposited in the European Nucleotide Archive under the accession number PRJEB38488. Other raw data are available from the Dryad Digital Repository: https://doi.org/10.5061/dryad.j9kd51cb9 [54].

**Authors' contributions.** M.B. and A.R.H. designed the study. M.B. performed experimental work and K.P.C. carried out the sequencing of the samples. M.B. and A.R.H. analysed the data and M.B., K.P.C. and A.R.H. wrote the manuscript. M.B., K.P.C. and A.R.H. revised the manuscript and approved the final version of the paper.

**Competing interests.** We declare we have no competing interests.

**Funding.** Funded by the Swiss National Science Foundation project 31003A_165803.

**Acknowledgements.** We thank the Genetic Diversity Center and Jean-Claude Walser for sequencing and bioinformatic support, and Pauline Scanlan for helpful feedback.

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
