## [Peer Review File · Proceedings of the Royal Society B: Biological Sciences]

Review History

RSPB-2020-2033.R0 (Original submission)

Review form: Reviewer 1

Recommendation

Major revision is needed (please make suggestions in comments)

Scientific importance: Is the manuscript an original and important contribution to its field?

Good

General interest: Is the paper of sufficient general interest?

Good

Quality of the paper: Is the overall quality of the paper suitable?

Acceptable

Is the length of the paper justified?

Yes

Should the paper be seen by a specialist statistical reviewer?

No

Do you have any concerns about statistical analyses in this paper? If so, please specify them explicitly in your report.

No

It is a condition of publication that authors make their supporting data, code and materials available - either as supplementary material or hosted in an external repository. Please rate, if applicable, the supporting data on the following criteria.

Is it accessible?

Yes

Is it clear?

Yes

Is it adequate?

Yes

Do you have any ethical concerns with this paper?

No

Comments to the Author

The authors pose an interesting question: What determines the ability of a gut community to prevent the establishment of an invasive species? They seek to disentangle the effect of the abiotic environment from the role of the taxonomic composition of communities. To that end, they use fecal derived communities in microcosms.

As I understand it, the authors show that *E. coli* is able to grow on its own in sterilized (autoclaved) fecal slurry, increasing in abundance by ~1 order of magnitude (Fig. 1 top panel). This growth is fast initially, but slower after 24 hours reaching an apparent stationary phase after 48hrs. When the invasive *E. coli* is added to a living community, it also increases in abundance at first (i.e. over the first 24hr or so), to levels comparable (if not higher) to those it reaches in monoculture (Fig. 1). Yet, over time its abundance declines. As these are total abundances (CFU/mL) this decline can only be attributed to cell death occurring faster than cell doublings in mixed culture relative to monoculture. The authors then show that *E. coli* can grow (though to a lesser extent) on the filtered supernatants of the communities, suggesting that resources have been depleted but also that the medium is not strictly toxic for *E. coli* (more on this later).

I commend the authors for the clarity of their writing and for addressing an interesting question. My main criticism is the difficulty to interpret their results. I would need to see the relative effect of cell death and births to fully be behind this paper. After reading the paper a few times, it is still unclear to me whether the suppressive effect they see on *E. coli* is due to increase cell death, decrease cell births or both.

If I make the assumption that cell death is negligible, then the growth to saturation observed in the first panel of Fig. 1 is due to cells reaching stationary phase. In such case, then I would expect to see in a live/dead stain that the fraction of live cells would be constant throughout the incubation and relatively small. I believe this (or another similar experiment) should be done to clarify the origin of stationary phase in monoculture.

Live/dead stain imaging would be challenging for the live communities, as the signal from the *E. coli* strain would be overwhelmed by the other community members. Perhaps the authors could take their monoculture at time 24hr, after which very little growth is observed, and add to it live cultures (rather than their supernatants) aged either 24 or 48hr (sterilized communities could be added as a control). If *E. coli* abundance declines afterwards, it would mark that cell death is being responsible for the decline. There may be other solutions to address this question, and I am

not necessarily imposing that one experiment. But some other way to clarify what is going on at a more mechanistic level would be necessary.

The supernatant experiment is interesting, but a bit unconvincing. The way this experiment was done, it does not rule out any contact dependent killing, nor does it eliminate the possibility of toxins that did not make it through the filtering process, or which were inactivated by it. I picture the slurry as a dense and physically non-homogeneous medium, which raises the question of whether certain particles (e.g. bacteriophages) may have not been unable to detach from this matrix to enter the supernatant.

The factorial experiment is also interesting, but again not mechanistically convincing.

I believe that the relative contributions of growth and death should be addressed in this paper before it is ready for publication. A few other minor comments are listed below:

Comment 1: In lines 74-84 the authors discuss resource competition as a mechanism to colonization resistance. They argue that this should be less sensitive to specific community composition, and more to the supplied resources. I agree with this interpretation with regard to competition in models. However, competition for resources in microbial communities can be more nuanced than this may suggest, as it often involves an interplay with facilitation. For instance, microbes routinely secrete metabolic byproducts, changing the environment as they do so and creating niches for others. They also secrete extracellular enzymes that break down complex matter, thereby making nutrients available for others. Altogether, cross-feeding and facilitation can create specific nutrient niches, which may affect whether or not a microorganism is able to invade. I would argue that, in this particular scenario, the relative importance of the nutrients that are supplied to the system and those that are created endogenously may determine whether colonization resistance is driven by which taxa are present (as this may dictate which endogenously generated nutrients are found) or by abiotic factors in the environment (the supplied nutrients). The authors should include this mechanism in their discussion.

Comment 2: Quantification of bacterial density using sybr green and flow cytometry is problematic, as not all cells will bind it equally. I am also not sure if any other particles in the fecal slurry may give false positives. The FSC and SSC are also very noisy for bacteria. The authors could add a quantification of the total amount of DNA to back their flow cytometry measurements? Since the authors have done 16 amplicon sequencing, they should have quantified the amount of DNA in their samples.

Review form: Reviewer 2

Recommendation

Accept with minor revision (please list in comments)

Scientific importance: Is the manuscript an original and important contribution to its field?

Good

General interest: Is the paper of sufficient general interest?

Good

Quality of the paper: Is the overall quality of the paper suitable?

Good

Is the length of the paper justified?

Yes

Should the paper be seen by a specialist statistical reviewer?

No

Do you have any concerns about statistical analyses in this paper? If so, please specify them explicitly in your report.

No

It is a condition of publication that authors make their supporting data, code and materials available - either as supplementary material or hosted in an external repository. Please rate, if applicable, the supporting data on the following criteria.

Is it accessible?

Yes

Is it clear?

Yes

Is it adequate?

Yes

Do you have any ethical concerns with this paper?

No

Comments to the Author

In this manuscript, the authors use a series of simple but well-designed experiments to examine the relative contribution of abiotic environment and community composition to colonization resistance within a model microbiome community. They find that complex microbiota are capable of resisting invasion by a model pathogen, but demonstrate that the extent of this invasion resistance depends upon the abiotic environment, likely nutrient limitation.

In general, I found this to be a well-written and interesting study. It's tempting to ask for further studies to more explicitly map out the mechanism driving this colonization resistance, but I think the paper as it stands already provides sufficient interesting material to merit publication, provided the following queries can be addressed:

Major comments:

Why was the supernatant experiment (Fig 4) aerobic, as opposed to anaerobic like all the other experiments? I worry when attempting to explain differences that one of these key experiments was performed under differing conditions. To a certain extent the point that environment alone doesn't explain the differing growth can be inferred from Fig 5, but if it is not an undue amount of work the manuscript would benefit from redoing Fig 4 to check this change in conditions does not impact the results.

Paragraph starting 309, the discussion of the community composition analysis, in particular whether or not shifts are driven by Enterobacteriaceae expansion, could do with more detail. Specifically, did the authors recalculate their relative abundances before redoing the PCoA analysis, or simply leave out the Enterobacteriaceae entry? I ask as if the latter, the information about Enterobacteriaceae expansion will still be present due to the compositional nature of the data.

Minor comments:

I found the section "Changing abiotic conditions alone do not explain focal strain suppression" a little hard to follow, and had to reread several times to establish exactly what had been done. This

is an important section, so I suggest rewording it slightly to improve clarity.

Figure 3, Enterobacteriaceae and Rikenellaceae are each colored the same, confusing things slightly. I suggest changing one.

If possible, I advise moving SI figure 1 into main text to help clarify your experimental procedure. Intro para starting line 62 could benefit from an explicit discussion that abiotic and biotic conditions are likely to influence one another.

Decision letter (RSPB-2020-2033.R0)

16-Sep-2020

Dear Dr Baumgartner:

I am writing to inform you that your manuscript RSPB-2020-2033 entitled "Microbial community composition interacts with local abiotic conditions to drive colonization resistance in human gut microbiome samples" has, in its current form, been rejected for publication in Proceedings B.

This action has been taken on the advice of referees, who have recommended that substantial revisions are necessary. With this in mind we would be happy to consider a resubmission, provided the comments of the referees are fully addressed. However please note that this is not a provisional acceptance.

Sincerely,
Professor Gary Carvalho
mailto: proceedingsb@royalsociety.org

Associate Editor

Board Member: 1

Comments to Author:

Your manuscript has been reviewed by two experts. Both note that the research questions are interesting and that the manuscript is well-written. Several critical points are raised about the work as currently presented. These relate to the level of mechanistic insight offered by the experiments (role of births vs deaths) and some important technical issues with some of the experimental designs (anaerobic vs aerobic culture) and analyses. If it is not possible to perform the requested additional experiments (for example due to covid restrictions in your country) please include appropriate statements explaining the limitations and caveats of the data.

Reviewer(s)' Comments to Author:

Referee: 1

Comments to the Author(s)

The authors pose an interesting question: What determines the ability of a gut community to prevent the establishment of an invasive species? They seek to disentangle the effect of the abiotic environment from the role of the taxonomic composition of communities. To that end, they use fecal derived communities in microcosms.

As I understand it, the authors show that *E. coli* is able to grow on its own in sterilized (autoclaved) fecal slurry, increasing in abundance by ~1 order of magnitude (Fig. 1 top panel). This growth is fast initially, but slower after 24 hours reaching an apparent stationary phase after 48hrs. When the invasive *E. coli* is added to a living community, it also increases in abundance at first (i.e. over the first 24hr or so), to levels comparable (if not higher) to those it reaches in monoculture (Fig. 1). Yet, over time its abundance declines. As these are total abundances (CFU/mL) this decline can only be attributed to cell death occurring faster than cell doublings in mixed culture relative to monoculture. The authors then show that *E. coli* can grow (though to a lesser extent) on the filtered supernatants of the communities, suggesting that resources have been depleted but also that the medium is not strictly toxic for *E. coli* (more on this later).

I commend the authors for the clarity of their writing and for addressing an interesting question. My main criticism is the difficulty to interpret their results. I would need to see the relative effect of cell death and births to fully be behind this paper. After reading the paper a few times, it is still unclear to me whether the suppressive effect they see on *E. coli* is due to increase cell death, decrease cell births or both.

If I make the assumption that cell death is negligible, then the growth to saturation observed in the first panel of Fig. 1 is due to cells reaching stationary phase. In such case, then I would expect to see in a live/dead stain that the fraction of live cells would be constant throughout the incubation and relatively small. I believe this (or another similar experiment) should be done to clarify the origin of stationary phase in monoculture.

Live/dead stain imaging would be challenging for the live communities, as the signal from the *E. coli* strain would be overwhelmed by the other community members. Perhaps the authors could take their monoculture at time 24hr, after which very little growth is observed, and add to it live cultures (rather than their supernatants) aged either 24 or 48hr (sterilized communities could be added as a control). If *E. coli* abundance declines afterwards, it would mark that cell death is being responsible for the decline. There may be other solutions to address this question, and I am not necessarily imposing that one experiment. But some other way to clarify what is going on at a more mechanistic level would be necessary.

The supernatant experiment is interesting, but a bit unconvincing. The way this experiment was done, it does not rule out any contact dependent killing, nor does it eliminate the possibility of toxins that did not make it through the filtering process, or which were inactivated by it. I picture the slurry as a dense and physically non-homogeneous medium, which raises the question of

whether certain particles (e.g. bacteriophages) may have not been unable to detach from this matrix to enter the supernatant.

The factorial experiment is also interesting, but again not mechanistically convincing.

I believe that the relative contributions of growth and death should be addressed in this paper before it is ready for publication. A few other minor comments are listed below:

Comment 1: In lines 74-84 the authors discuss resource competition as a mechanism to colonization resistance. They argue that this should be less sensitive to specific community composition, and more to the supplied resources. I agree with this interpretation with regard to competition in models. However, competition for resources in microbial communities can be more nuanced than this may suggest, as it often involves an interplay with facilitation. For instance, microbes routinely secrete metabolic byproducts, changing the environment as they do so and creating niches for others. They also secrete extracellular enzymes that break down complex matter, thereby making nutrients available for others. Altogether, cross-feeding and facilitation can create specific nutrient niches, which may affect whether or not a microorganism is able to invade. I would argue that, in this particular scenario, the relative importance of the nutrients that are supplied to the system and those that are created endogenously may determine whether colonization resistance is driven by which taxa are present (as this may dictate which endogenously generated nutrients are found) or by abiotic factors in the environment (the supplied nutrients). The authors should include this mechanism in their discussion.

Comment 2: Quantification of bacterial density using sybr green and flow cytometry is problematic, as not all cells will bind it equally. I am also not sure if any other particles in the fecal slurry may give false positives. The FSC and SSC are also very noisy for bacteria. The authors could add a quantification of the total amount of DNA to back their flow cytometry measurements? Since the authors have done 16 amplicon sequencing, they should have quantified the amount of DNA in their samples.

Referee: 2

Comments to the Author(s)

In this manuscript, the authors use a series of simple but well-designed experiments to examine the relative contribution of abiotic environment and community composition to colonization resistance within a model microbiome community. They find that complex microbiota are capable of resisting invasion by a model pathogen, but demonstrate that the extent of this invasion resistance depends upon the abiotic environment, likely nutrient limitation.

In general, I found this to be a well-written and interesting study. It's tempting to ask for further studies to more explicitly map out the mechanism driving this colonization resistance, but I think the paper as it stands already provides sufficient interesting material to merit publication, provided the following queries can be addressed:

Major comments:

Why was the supernatant experiment (Fig 4) aerobic, as opposed to anaerobic like all the other experiments? I worry when attempting to explain differences that one of these key experiments was performed under differing conditions. To a certain extent the point that environment alone doesn't explain the differing growth can be inferred from Fig 5, but if it is not an undue amount of work the manuscript would benefit from redoing Fig 4 to check this change in conditions does not impact the results.

Paragraph starting 309, the discussion of the community composition analysis, in particular whether or not shifts are driven by Enterobacteriaceae expansion, could do with more detail.

Specifically, did the authors recalculate their relative abundances before redoing the PCoA analysis, or simply leave out the Enterobacteriaceae entry? I ask as if the latter, the information about Enterobacteriaceae expansion will still be present due to the compositional nature of the data.

Minor comments:

I found the section “Changing abiotic conditions alone do not explain focal strain suppression” a little hard to follow, and had to reread several times to establish exactly what had been done. This is an important section, so I suggest rewording it slightly to improve clarity.

Figure 3, Enterobacteriaceae and Rikenellaceae are each colored the same, confusing things slightly. I suggest changing one.

If possible, I advise moving SI figure 1 into main text to help clarify your experimental procedure

Intro para starting line 62 could benefit from an explicit discussion that abiotic and biotic conditions are likely to influence one another.

Author's Response to Decision Letter for (RSPB-2020-2033.R0)

See Appendix A.

RSPB-2020-3106.R0

Review form: Reviewer 2

Recommendation

Accept as is

Scientific importance: Is the manuscript an original and important contribution to its field?

Good

General interest: Is the paper of sufficient general interest?

Good

Quality of the paper: Is the overall quality of the paper suitable?

Good

Is the length of the paper justified?

Yes

Should the paper be seen by a specialist statistical reviewer?

No

Do you have any concerns about statistical analyses in this paper? If so, please specify them explicitly in your report.

No

It is a condition of publication that authors make their supporting data, code and materials available - either as supplementary material or hosted in an external repository. Please rate, if applicable, the supporting data on the following criteria.

Is it accessible?

Yes

Is it clear?

Yes

Is it adequate?

Yes

Do you have any ethical concerns with this paper?

No

Comments to the Author

I thank the authors for clearly addressing each of my concerns. While I agree with Referee 1 that it would be interesting to get a better view of what is driving the colonization resistance observed, I think as it stands this work already provides a useful contribution to the field and merits being published here as is. I do still think if space allows the manuscript would benefit from including the schematic in the main text, but I appreciate this is a matter of taste.

Review form: Reviewer 3

Recommendation

Accept with minor revision (please list in comments)

Scientific importance: Is the manuscript an original and important contribution to its field?

Good

General interest: Is the paper of sufficient general interest?

Excellent

Quality of the paper: Is the overall quality of the paper suitable?

Good

Is the length of the paper justified?

Yes

Should the paper be seen by a specialist statistical reviewer?

No

Do you have any concerns about statistical analyses in this paper? If so, please specify them explicitly in your report.

Yes

It is a condition of publication that authors make their supporting data, code and materials available - either as supplementary material or hosted in an external repository. Please rate, if applicable, the supporting data on the following criteria.

Is it accessible?

Yes

Is it clear?

Yes

Is it adequate?

N/A

Do you have any ethical concerns with this paper?

No

Comments to the Author

Major comments

Through-out the text you talk about “adaptation” of the community and how this affects invasion success. Reading the text my impression is what you have in mind is adjustment of the community performance by ecological means, like species sorting or ecological drift. I don't think you mean adaptive evolution in terms of heritable modifications of the phenotypes through genetic changes. As you maintained your cultures for 72h, it may be a reasonable assumption that mainly ecological forces drive community change. If this is true, your terminology through-out the text is misleading. In case you indeed think adaptive evolution occurred in your experiment, I would be more critical asking for proper measures how much fitness of particular species changed and how much your focal species evolved. While I agree that within 72h evolution may not be very dominant, I don't think it can be entirely excluded! We have good indication that bacteria can rapidly evolve even within a few generations and that this can have effects on community functions. Not looking for it doesn't mean it didn't happen! This should be at least discussed and we have recent literature exploring evolution in complex bacterial communities. I see this was a decent amount of work so I wouldn't straight ask for additional experiments. If you are sure no evolution occurred, you probably can argue out of that. How many generations you think happened during these 72h hours? When I went through the text the first time, I thought you calculated a generation number; reading it again I was unable to spot it.

Minor comments

L48: Well, it may also depend on chance, initial frequencies (is this a single individual of the invader, or a large population) and the ability for rapid evolution!

L64: I would assume that high diversity isn't the sole driver, but how well the resident community capitalizes resources needed by the invader, which of course is a function of biodiversity. This goes into the direction of functional redundancy in microbial communities. With high levels of redundancy, invasion should be more difficult compared to strong niche partition. This now should also interact with available resources. I would see a close interaction between the invading strain properties and the resident community...

L71: However, when resources become scarce, competitive power of resident communities may also decrease to outcompete the invader (reduced cell densities). I guess there are two sides of the coin. Maybe represent these two less biased.

L73: There are several well established examples (Goldford Science 2018, Lawrence PlosBio 2012, Fiegna ProcB 2015) that cross feeding has effects on coexistence.

L 134: It would be helpful to briefly describe how “Hungate” tubes work. While there is a reference this “abiotic” condition is crucial for the study question.

L 137: What was the CFU of the live slurry?

L145-148: There is an extrapolation from total cell density measured by flow cytometry towards ratio of the invader measured by plating. As I understand it, the community density was not determined by plating. I can see that double plating would be too much work. Are 100% of the

invader cells surviving the plating process? In the ideal case there should be a 1:1 line, cell counts from the flow and the CFU, which may be a point in the Supplements.

L170: Not sure if the word “adapted” is appropriate. I guess the authors don’t use it in an evolutionary context but mean the communities have ecologically stabilized? But see comments on evolution....

L173: I didn’t really understand how the swapping was done. Supp Fig. S1 is not very informative on this. Maybe redraw the figure or give a bit more info in the text. There is more information in the Supplementary Methods but still it doesn’t become clear to me. The wording “fresh community+fresh” and “adapted community+spend” is also confusing. Is there also fresh community+spend? When you talk about “adapted community”, that has been cultured for 72h right? “Fresh community” would then be grown for 24 h? Well, maybe this becomes clearer in the Results...

L178: When you say “after 72h” you mean the 72h grown cultures, right?

L199: I guess a specific region of the 16S gene was sequenced? Ah given in the Supplements... Why not just adding it in the Methods?

L220: By the way fresh medium is unused medium, right? So, you added the bacterial community and directly removed it?

L211: What is the difference between “Donor and Community” in the fixed effects of you model. The community was derived from the donors, right? I guess you have some nasty correlation situation where you track Community over Time but Community is correlated with Donor... While “Replicate” and “ID” will address the temporal correlation, I guess you need either additional random terms telling the model community is nested in Donor or correlation structures to address the non-independency between Community and Donor. Maybe nlme is better equipped to include the correlation structure.

L236: The lmer will give you a slope over time! When you say the effect was strongest towards the end, where does this come from? Did you include non-linear trends indicating a change over time? Or have you used “time” as a factor? In this case you would get differences for time points compared to the starting time point.

L243: According with my comment above, can it be excluded that donor1 and community1 are independent from each other? With donor1 you refer to slurry from donor1, right?

L249: Would this mean there was active killing by the community? If the focal strain density wouldn’t increase but the community would grow I would think competition constrains the invader but there is no active killing. Of course, here we run into the problem that the community may simply grow at a grow/death balance, which is likely hard to exclude...

L263: A bit of poor luck that you invader was hitting the ecological dominant species in your experiments. Now this interacts with my comment above. Resident Ecoli are increasing while invading Ecoli die, is there active killing or sever competition?

L278: This also indicates non-independency between community and donor...

L306: Guess you mean compared to control rather than sterile slurry.

L315: In line 310 you say “environment became NOT nutrient poor”. This is a bit confusing. It became poorer but not so much it would prevent growth. Maybe re-word a bit.

L326: and you grew them for XX days? I mean your fresh community would become adapted after 72 hours...

L337: I am not sure you really "showed" that colonization success depends on interaction. I feel you have good indication but can also see alternative explanations. Neither it is clear "demonstration" L339...

L343: Have you really convincing indication that nutrients get depleted? It could be more an exhaustion of easily accessible resources with still much carbon for growth available bound in recalcitrant form.

L361: "Less diverse communities seem to be harder to invade". This connects with my main criticism of this work. When members of the community evolved, in terms of fitness change and resource use, more complete pre-emption of niches would constrain invaders even when non-functional diversity is lost. Are your communities going through adaptive evolution or is this some "community adaptation process" by means of species sorting and ecological drift?

L369: What else would we need? Just remind the reader!

L382: You should compare your findings to previous results from the human microbiome consortium. There, host diet and physiological status were also discussed.

Figures:

Fig. 1 These are means of 3 replicates, right? Maybe add error bars!

So, this means you added communities 1-3 to sterile slurry from donor 1 and similarly to slurries from d2-d3? This also only became clear to me after looking into the figure... This now goes back into my comment on stats what variables are donor & community...

What I am somehow missing is how communities did in "Control" medium. You didn't do this, right?

Fig2: It is good that communities don't die and interesting to see they don't grow biomass anymore. Anything else we learn from the figure? If that figure would go into the supplements I wouldn't miss it...

Fig. 3: "Adapted 1-3" means three replicates from last time point, right? Or is this pooled reps for 3 time points?

Fig. 4: This is final density after 24hours? Correct me, but in the control medium nutrients were not pre-used (unless the invader was grown in it) but the slurry is metabolized already when you got it from the donor. I guess control supernatant from communities would be helpful... In general, we learn from this that the slurry still has lots of nutrients inside and the additional 72 hours don't change this much. Have 72 hours been enough to run the experiment when growth is obviously slow?

Review form: Reviewer 4

Recommendation

Accept with minor revision (please list in comments)

Scientific importance: Is the manuscript an original and important contribution to its field?

Good

General interest: Is the paper of sufficient general interest?

Excellent

Quality of the paper: Is the overall quality of the paper suitable?

Excellent

Is the length of the paper justified?

Yes

Should the paper be seen by a specialist statistical reviewer?

No

Do you have any concerns about statistical analyses in this paper? If so, please specify them explicitly in your report.

No

It is a condition of publication that authors make their supporting data, code and materials available - either as supplementary material or hosted in an external repository. Please rate, if applicable, the supporting data on the following criteria.

Is it accessible?

Yes

Is it clear?

Yes

Is it adequate?

Yes

Do you have any ethical concerns with this paper?

No

Comments to the Author

In their manuscript "Microbial community composition interacts with local abiotic conditions to drive colonization resistance in human gut microbiome samples" Baumgartner et al. elucidate how biotic and abiotic factors shape the resilience of human gut microbial communities against invasion through a focal *E. coli* strain. It hence provides novel insights into an important question associated to human health.

While a clear mechanistic understanding into how exactly resilience of microbial communities is shaped is lacking and outside the scope of this study, the manuscript provides a great basis to explore the hypothesized mechanisms in depth in follow up studies.

The manuscript is well written, easy to understand and logically structured. The authors rely on a set of simple, but very well designed experimental assays that built upon each other to increase the complexity of tested biotic and abiotic variables.

The authors have replied well and sufficiently to the comments raised by the reviewers and consequently improved the quality of the manuscript. I believe that there are only a few minor remaining points to address.

L. 23: Rephrase: Community composition was only measured after 0 and 72h. As dynamics in composition shifts were not monitored across this period, it is not possible to associate shifts in community composition to later time points exclusively.

L. 326-328: rephrase, to ensure the clear statement of this important result

Fig. 3: Ensure capitalization of "Human Donor" and "(Community, Replicate)" is consistent between panels.

Decision letter (RSPB-2020-3106.R0)

01-Feb-2021

Dear Dr Baumgartner:

Your manuscript has now been peer reviewed and the reviews have been assessed by an Associate Editor. The reviewers' comments (not including confidential comments to the Editor) and the comments from the Associate Editor are included at the end of this email for your reference. As you will see, the reviewers and the Editors have raised some concerns with your manuscript and we would like to invite you to revise your manuscript to address them.

As you will know, an important feature of PRSB, is to maximise accessibility and understanding across a broad readership. In that vein, I endorse the comments and suggestion from referee one, concerning the value of an overall schematic, and encourage you strongly to consider including one.

Research ethics:

Use of animals and field studies:

If your study uses animals please include details in the methods section of any approval and licences given to carry out the study and include full details of how animal welfare standards

were ensured. Field studies should be conducted in accordance with local legislation; please include details of the appropriate permission and licences that you obtained to carry out the field work.

It is a condition of publication that you make available the data and research materials supporting the results in the article (<https://royalsociety.org/journals/authors/author-guidelines/#data>). Datasets should be deposited in an appropriate publicly available repository and details of the associated accession number, link or DOI to the datasets must be included in the Data Accessibility section of the article (<https://royalsociety.org/journals/ethics-policies/data-sharing-mining/>). Reference(s) to datasets should also be included in the reference list of the article with DOIs (where available).

Please submit a copy of your revised paper within three weeks. If we do not hear from you within this time your manuscript will be rejected. If you are unable to meet this deadline please let us know as soon as possible, as we may be able to grant a short extension.

Best wishes,
Professor Gary Carvalho
<mailto:proceedingsb@royalsociety.org>

Associate Editor

Comments to Author:

Thank you for resubmitting your revised manuscript. This was sent to one of the original reviewers and 2 new reviewers. As you will see the original reviewer is entirely satisfied, and the 2 new reviewers raise a few minor issues that can be addressed in revision.

Reviewer(s)' Comments to Author:

Referee: 2

Comments to the Author(s).

I thank the authors for clearly addressing each of my concerns. While I agree with Referee 1 that it would be interesting to get a better view of what is driving the colonization resistance observed, I think as it stands this work already provides a useful contribution to the field and merits being published here as is. I do still think if space allows the manuscript would benefit from including the schematic in the main text, but I appreciate this is a matter of taste.

Referee: 3

Comments to the Author(s).

Major comments

Through-out the text you talk about “adaptation” of the community and how this affects invasion success. Reading the text my impression is what you have in mind is adjustment of the community performance by ecological means, like species sorting or ecological drift. I don't think you mean adaptive evolution in terms of heritable modifications of the phenotypes through genetic changes. As you maintained your cultures for 72h, it may be a reasonable assumption that mainly ecological forces drive community change. If this is true, your terminology through-out the text is misleading. In case you indeed think adaptive evolution occurred in your experiment, I would be more critical asking for proper measures how much fitness of particular species changed and how much your focal species evolved. While I agree that within 72h evolution may not be very dominant, I don't think it can be entirely excluded! We have good indication that bacteria can rapidly evolve even within a few generations and that this can have effects on community functions. Not looking for it doesn't mean it didn't happen! This should be at least discussed and we have recent literature exploring evolution in complex bacterial communities. I see this was a decent amount of work so I wouldn't straight ask for additional experiments. If you are sure no evolution occurred, you probably can argue out of that. How many generations you think happened during these 72h hours? When I went through the text the first time, I thought you calculated a generation number; reading it again I was unable to spot it.

Minor comments

L48: Well, it may also depend on chance, initial frequencies (is this a single individual of the invader, or a large population) and the ability for rapid evolution!

L64: I would assume that high diversity isn't the sole driver, but how well the resident community capitalizes resources needed by the invader, which of course is a function of biodiversity. This goes into the direction of functional redundancy in microbial communities. With high levels of redundancy, invasion should be more difficult compared to strong niche partition. This now should also interact with available resources. I would see a close interaction between the invading strain properties and the resident community...

L71: However, when resources become scarce, competitive power of resident communities may also decrease to outcompete the invader (reduced cell densities). I guess there are two sides of the coin. Maybe represent these two less biased.

L73: There are several well established examples (Goldford Science 2018, Lawrence PlosBio 2012, Fiegna ProcB 2015) that cross feeding has effects on coexistence.

L 134: It would be helpful to briefly describe how “Hungate” tubes work. While there is a reference this “abiotic” condition is crucial for the study question.

L 137: What was the CFU of the live slurry?

L145-148: There is an extrapolation from total cell density measured by flow cytometry towards ratio of the invader measured by plating. As I understand it, the community density was not

determined by plating. I can see that double plating would be too much work. Are 100% of the invader cells surviving the plating process? In the ideal case there should be a 1:1 line, cell counts from the flow and the CFU, which may be a point in the Supplements.

L170: Not sure if the word “adapted” is appropriate. I guess the authors don’t use it in an evolutionary context but mean the communities have ecologically stabilized? But see comments on evolution....

L173: I didn’t really understand how the swapping was done. Supp Fig. S1 is not very informative on this. Maybe redraw the figure or give a bit more info in the text. There is more information in the Supplementary Methods but still it doesn’t become clear to me. The wording “fresh community+fresh” and “adapted community+spend” is also confusing. Is there also fresh community+spend? When you talk about “adapted community”, that has been cultured for 72h right? “Fresh community” would then be grown for 24 h? Well, maybe this becomes clearer in the Results...

L178: When you say “after 72h” you mean the 72h grown cultures, right?

L199: I guess a specific region of the 16S gene was sequenced? Ah given in the Supplements... Why not just adding it in the Methods?

L220: By the way fresh medium is unused medium, right? So, you added the bacterial community and directly removed it?

L211: What is the difference between “Donor and Community” in the fixed effects of your model. The community was derived from the donors, right? I guess you have some nasty correlation situation where you track Community over Time but Community is correlated with Donor... While “Replicate” and “ID” will address the temporal correlation, I guess you need either additional random terms telling the model community is nested in Donor or correlation structures to address the non-independency between Community and Donor. Maybe nlme is better equipped to include the correlation structure.

L236: The lmer will give you a slope over time! When you say the effect was strongest towards the end, where does this come from? Did you include non-linear trends indicating a change over time? Or have you used “time” as a factor? In this case you would get differences for time points compared to the starting time point.

L243: According with my comment above, can it be excluded that donor1 and community1 are independent from each other? With donor1 you refer to slurry from donor1, right?

L249: Would this mean there was active killing by the community? If the focal strain density wouldn’t increase but the community would grow I would think competition constrains the invader but there is no active killing. Of course, here we run into the problem that the community may simply grow at a grow/death balance, which is likely hard to exclude...

L263: A bit of poor luck that your invader was hitting the ecological dominant species in your experiments. Now this interacts with my comment above. Resident Ecoli are increasing while invading Ecoli die, is there active killing or severe competition?

L278: This also indicates non-independency between community and donor...

L306: Guess you mean compared to control rather than sterile slurry.

L315: In line 310 you say “environment became NOT nutrient poor”. This is a bit confusing. It became poorer but not so much it would prevent growth. Maybe re-word a bit.

L326: and you grew them for XX days? I mean your fresh community would become adapted after 72 hours...

L337: I am not sure you really “showed” that colonization success depends on interaction. I feel you have good indication but can also see alternative explanations. Neither it is clear “demonstration” L339...

L343: Have you really convincing indication that nutrients get depleted? It could be more an exhaustion of easily accessible resources with still much carbon for growth available bound in recalcitrant form.

L361: “Less diverse communities seem to be harder to invade”. This connects with my main criticism of this work. When members of the community evolved, in terms of fitness change and resource use, more complete pre-emption of niches would constrain invaders even when non-functional diversity is lost. Are your communities going through adaptive evolution or is this some “community adaptation process” by means of species sorting and ecological drift?

L369: What else would we need? Just remind the reader!

L382: You should compare your findings to previous results from the human microbiome consortium. There, host diet and physiological status were also discussed.

Figures:

Fig. 1 These are means of 3 replicates, right? Maybe add error bars!

So, this means you added communities 1-3 to sterile slurry from donor 1 and similarly to slurries from d2-d3? This also only became clear to me after looking into the figure... This now goes back into my comment on stats what variables are donor & community...

What I am somehow missing is how communities did in “Control” medium. You didn’t do this, right?

Fig2: It is good that communities don’t die and interesting to see they don’t grow biomass anymore. Anything else we learn from the figure? If that figure would go into the supplements I wouldn’t miss it...

Fig. 3: “Adapted 1-3” means three replicates from last time point, right? Or is this pooled reps for 3 time points?

Fig. 4: This is final density after 24hours? Correct me, but in the control medium nutrients were not pre-used (unless the invader was grown in it) but the slurry is metabolized already when you got it from the donor. I guess control supernatant from communities would be helpful... In general, we learn from this that the slurry still has lots of nutrients inside and the additional 72 hours don’t change this much. Have 72 hours been enough to run the experiment when growth is obviously slow?

Referee: 4

Comments to the Author(s).

In their manuscript “Microbial community composition interacts with local abiotic conditions to drive colonization resistance in human gut microbiome samples” Baumgartner et al. elucidate how biotic and abiotic factors shape the resilience of human gut microbial communities against invasion through a focal *E. coli* strain. It hence provides novel insights into an important question associated to human health.

While a clear mechanistic understanding into how exactly resilience of microbial communities is shaped is lacking and outside the scope of this study, the manuscript provides a great basis to explore the hypothesized mechanisms in depth in follow up studies.

The manuscript is well written, easy to understand and logically structured. The authors rely on a set of simple, but very well designed experimental assays that built upon each other to increase the complexity of tested biotic and abiotic variables.

The authors have replied well and sufficiently to the comments raised by the reviewers and consequently improved the quality of the manuscript. I believe that there are only a few minor remaining points to address.

L. 23: Rephrase: Community composition was only measured after 0 and 72h. As dynamics in composition shifts were not monitored across this period, it is not possible to associate shifts in community composition to later time points exclusively.

L. 326-328: rephrase, to ensure the clear statement of this important result

Fig. 3: Ensure capitalization of "Human Donor" and "(Community, Replicate)" is consistent between panels.

Author's Response to Decision Letter for (RSPB-2020-3106.R0)

See Appendix B.

Decision letter (RSPB-2020-3106.R1)

25-Feb-2021

Dear Dr Baumgartner

I am pleased to inform you that your manuscript entitled "Microbial community composition interacts with local abiotic conditions to drive colonization resistance in human gut microbiome samples" has been accepted for publication in Proceedings B.

Open Access

Paper charges

Sincerely,

Professor Gary Carvalho

Appendix A

Associate Editor

Board Member: 1

Comments to Author:

Your manuscript has been reviewed by two experts. Both note that the research questions are interesting and that the manuscript is well-written. Several critical points are raised about the work as currently presented. These relate to the level of mechanistic insight offered by the experiments (role of births vs deaths) and some important technical issues with some of the experimental designs (anaerobic vs aerobic culture) and analyses. If it is not possible to perform the requested additional experiments (for example due to covid restrictions in your country) please include appropriate statements explaining the limitations and caveats of the data.

We detail our responses to individual comments below. In summary: (1) we carried out a new experiment to address the reviewer's concern about part of the study being aerobic rather than anaerobic, (2) we better defined the scope of the paper, including discussion of the roles of births vs deaths that one reviewer raised, (3) we carried out a small additional experiment to confirm that our fluorescence flow cytometry method for detecting bacteria does not pick up significant background noise.

Although we were able to do these two relatively straightforward additional experiments, we were unable to add further experiments about the roles of births and deaths. To do this properly would require more work than is feasible in the six-month window for resubmission, particularly because our anaerobic work requires access to a laboratory in another department, and this is restricted by corona-related measures at our university. The set-up and troubleshooting that would be required for meaningful experiments on births vs deaths in this system make it unrealistic. Moreover, as we now say in the manuscript, while we agree with the reviewer that this issue is interesting, it is not central to the questions we addressed, interpretation of our results, or their main impact and novelty. We hope the revised manuscript therefore addresses the reviewers' concerns and makes the scope, limitations and impact of the paper clearer.

Reviewer(s)' Comments to Author:

Referee: 1

Comments to the Author(s)

The authors pose an interesting question: What determines the ability of a gut community to prevent the establishment of an invasive species? They seek to disentangle the effect of the abiotic environment from the role of the taxonomic composition of communities. To that end, they use fecal derived communities in microcosms.

*As I understand it, the authors show that *E. coli* is able to grow on its own in sterilized (autoclaved) fecal slurry, increasing in abundance by ~1 order of magnitude (Fig. 1 top panel). This growth is fast initially, but slower after 24 hours reaching an apparent stationary phase after 48hrs. When the invasive *E. coli* is added to a living community, it also increases in abundance at first (i.e. over the first 24hr or so), to levels comparable (if not higher) to those it reaches in monoculture (Fig. 1). Yet, over time its abundance declines. As these are total abundances (CFU/mL) this decline can only be attributed to cell death occurring faster than cell doublings in mixed culture relative to monoculture. The authors then show that *E. coli* can grow (though to a lesser extent) on the filtered supernatants of the communities, suggesting that resources have been depleted but also that the medium is not strictly toxic for *E. coli* (more on this later).*

*I commend the authors for the clarity of their writing and for addressing an interesting question. My main criticism is the difficulty to interpret their results. I would need to see the relative effect of cell death and births to fully be behind this paper. After reading the paper a few times, it is still unclear to me whether the suppressive effect they see on *E. coli* is due to increase cell death, decrease cell births or both.*

It is indeed hard to resolve the changes in birth vs death rates from our data. We now point out this limitation in the discussion (L400-406). We also agree this is an interesting question which would provide insight into how communities affect invaders, and we have now mentioned this in the revised introduction (L78-82). Nevertheless, the central question we addressed here is about the effect of

communities on invasion success, which ultimately depends on net population growth of the invading strain/species. We therefore think that our key finding, that community composition and the abiotic environment interact to determine invasion success in human microbiota samples (going beyond past work with, for example, murine model systems), still provides a novel contribution to both our basic understanding of how communities respond to invasion and our ability to manipulate this in microbiota-based therapy, even without breaking it down to births and deaths. We have edited the text to explain this in the introduction (L82-91), and the limitations/scope of our study in this context in the discussion (400-406).

If I make the assumption that cell death is negligible, then the growth to saturation observed in the first panel of Fig. 1 is due to cells reaching stationary phase. In such case, then I would expect to see in a live/dead stain that the fraction of live cells would be constant throughout the incubation and relatively small. I believe this (or another similar experiment) should be done to clarify the origin of stationary phase in monoculture.

Live/dead stain imaging would be challenging for the live communities, as the signal from the E. coli strain would be overwhelmed by the other community members. Perhaps the authors could take their monoculture at time 24hr, after which very little growth is observed, and add to it live cultures (rather than their supernatants) aged either 24 or 48hr (sterilized communities could be added as a control). If E. coli abundance declines afterwards, it would mark that cell death is being responsible for the decline. There may be other solutions to address this question, and I am not necessarily imposing that one experiment. But some other way to clarify what is going on at a more mechanistic level would be necessary.

This is an interesting suggestion, although we would be concerned that increasing the abundance of viable cells way above their stationary phase density could lead to death rates that do not reflect those in the main experiment. Moreover, as explained above, it is not feasible for us to develop such experiments during the timeframe for submission, particularly because of covid-related restrictions, and we do not think this issue is essential for interpretation of our key findings (L351 and L374). However, we did investigate possible ways of tracking births/deaths in such conditions, and identified two possible strategies that might be pursued in future work. We added reference to these in the discussion (L404-406).

The supernatant experiment is interesting, but a bit unconvincing. The way this experiment was done, it does not rule out any contact dependent killing, nor does it eliminate the possibility of toxins that did not make it through the filtering process, or which were inactivated by it. I picture the slurry as a dense and physically non-homogeneous medium, which raises the question of whether certain particles (e.g. bacteriophages) may have not been able to detach from this matrix to enter the supernatant.

The slurry in the microcosms is more like a liquid/suspension, because faecal samples are homogenized in peptone wash and further diluted. It is nevertheless true that, despite homogenization and vortexing, some fraction of the matrix might remain intact/attached. We added discussion of this limitation, including how it could contribute to the weaker negative effects in supernatant compared to in live slurry (L421-424). We also address the possibility of bacteriophages in the supernatant, and why we do not think this influenced our results, in the discussion (L412-L421).

The factorial experiment is also interesting, but again not mechanistically convincing.

I believe that the relative contributions of growth and death should be addressed in this paper before it is ready for publication. A few other minor comments are listed below:

As outlined above, we hope the revised introduction (L75-91) and discussion (L400-406) integrate the births-vs-deaths issue into our interpretation, better explain why this is not central to our core findings, and point to possible ways of addressing this in future work.

Comment 1: In lines 74-84 the authors discuss resource competition as a mechanism to colonization resistance. They argue that this should be less sensitive to specific community composition, and more to the supplied resources. I agree with this interpretation with regard to competition in models. However, competition for resources in microbial communities can be more nuanced than this may suggest, as it often involves an interplay with facilitation. For instance, microbes routinely secrete metabolic byproducts, changing the environment as they do so and creating niches for others. They

also secrete extracellular enzymes that break down complex matter, thereby making nutrients available for others. Altogether, cross-feeding and facilitation can create specific nutrient niches, which may affect whether or not a microorganism is able to invade. I would argue that, in this particular scenario, the relative importance of the nutrients that are supplied to the system and those that are created endogenously may determine whether colonization resistance is driven by which taxa are present (as this may dictate which endogenously generated nutrients are found) or by abiotic factors in the environment (the supplied nutrients). The authors should include this mechanism in their discussion.

This is a very interesting point, we have incorporated it into the introduction (L71-73) as suggested.

Comment 2: Quantification of bacterial density using sybr green and flow cytometry is problematic, as not all cells will bind it equally. I am also not sure if any other particles in the fecal slurry may give false positives. The FSC and SSC are also very noisy for bacteria. The authors could add a quantification of the total amount of DNA to back their flow cytometry measurements? Since the authors have done 16 amplicon sequencing, they should have quantified the amount of DNA in their samples.

SYBR green is widely used in flow cytometric enumeration of bacterial populations from different environments. We established the settings we used to discriminate cells from background noise after extensive preliminary work. However, in response to this comment, we have further validated our method by measuring basal growth medium and sterilized faecal slurry (accounting for possible false positives from nonviable cells in faecal slurry), and with/without added focal strain (confirming the focal strain signal is detectable above any background noise). As shown in the cytograms below, this demonstrates negligible background signal in treatments without the focal strain (focal strain is

labelled "Bacteria" here; we record hits only inside the red box), and that the viable focal strain is clearly distinguishable from debris/nonviable cells in the sterilized slurry. However, we agree that flow cytometric measurements have other limitations (e.g., cell aggregates counted as a single event). While we do not expect this to have influenced our analysis of relative bacterial abundances over time, we nevertheless added a mention of these limitations in the methods section (L151-153).

We acknowledge that total DNA might be a good indicator for biomass in some contexts, but would prefer not to use it as a proxy for cell density here for the following reasons. Considering the range of bacterial genome sizes, the total DNA yield will depend on community structure, and this might obscure our estimation of population density. Furthermore, DNA extraction is not equally efficient for each cell type, which might again interfere with using total DNA concentration as a proxy for total cell counts.

Referee: 2

Comments to the Author(s)

In this manuscript, the authors use a series of simple but well-designed experiments to examine the relative contribution of abiotic environment and community composition to colonization resistance within a model microbiome community. They find that complex microbiota are capable of resisting invasion by a model pathogen, but demonstrate that the extent of this invasion resistance depends upon the abiotic environment, likely nutrient limitation.

In general, I found this to be a well-written and interesting study. It's tempting to ask for further studies to more explicitly map out the mechanism driving this colonization resistance, but I think the paper as it stands already provides sufficient interesting material to merit publication, provided the following queries can be addressed:

Major comments:

Why was the supernatant experiment (Fig 4) aerobic, as opposed to anaerobic like all the other experiments? I worry when attempting to explain differences that one of these key experiments was performed under differing conditions. To a certain extent the point that environment alone doesn't explain the differing growth can be inferred from Fig 5, but if it is not an undue amount of work the manuscript would benefit from redoing Fig 4 to check this change in conditions does not impact the results.

We agree this was a shortcoming of the previous version. We have now repeated the supernatant experiment under anaerobic conditions, using a similar approach as in the experiment behind Figure 5. The updated figure supports an extremely similar pattern to the previous version, and is shown in Figure 4. We have also updated the corresponding methods section (L162-167).

Paragraph starting 309, the discussion of the community composition analysis, in particular whether or not shifts are driven by Enterobacteriaceae expansion, could do with more detail. Specifically, did the authors recalculate their relative abundances before redoing the PCoA analysis, or simply leave out the Enterobacteriaceae entry? I ask as if the latter, the information about Enterobacteriaceae expansion will still be present due to the compositional nature of the data.

Thank you for pointing this out. Indeed, we previously did not recalculate relative abundances after removing Enterobacteriaceae. We repeated the analysis by including this step and substituted the figure (Sup. Figure 1B); the conclusions are unchanged (L278-282 in main text, where we also mention explicitly that we recalculated abundances).

Minor comments:

I found the section "Changing abiotic conditions alone do not explain focal strain suppression" a little hard to follow, and had to reread several times to establish exactly what had been done. This is an important section, so I suggest rewording it slightly to improve clarity.

We have rephrased this section to clarify what we did and what we measured.

Figure 3, Enterobacteriaceae and Rikenellaceae are each colored the same, confusing things slightly. I

suggest changing one.

Thanks for spotting this. We changed the colour scheme.

If possible, I advise moving SI figure 1 into main text to help clarify your experimental procedure

We tried various configurations here, but in the end prefer to keep it as supplementary to avoid adding significantly to the length of the main text (which we estimate to be already approaching the 10-page limit). Instead, we added more direct reference to the schematic in the corresponding section of the main text, so any readers who need further illustration are directed straight to the schematic (L324 + L326).

Intro para starting line 62 could benefit from an explicit discussion that abiotic and biotic conditions are likely to influence one another.

We expanded on this in the introduction (L75-L91).

Appendix B

Zurich, 20.02.21

Resubmission of manuscript RSPB-2020-3106 entitled "Microbial community composition interacts with local abiotic conditions to drive colonization resistance in human gut microbiome samples"

Dear Prof Carvalho and Associate Editor,

We thank you and the reviewers for suggestions that helped us further improve the manuscript. In summary, we addressed all comments by: (1) adding clarifications to the text and an overview panel to our schematic figure S1, (2) refining some of our explanations of experimental design and analysis, (3) updating some of the references and discussion points as directed by the reviewers. We provide details of each response below; reviewer and editor comments are in blue, our responses in black. We hope you find the revised manuscript further improved and will consider it for publication in Proceedings B.

Sincerely,

Michael Baumgartner (on behalf of all authors)

Dear Dr Baumgartner:

Your manuscript has now been peer reviewed and the reviews have been assessed by an Associate Editor. The reviewers' comments (not including confidential comments to the Editor) and the comments from the Associate Editor are included at the end of this email for your reference. As you will see, the reviewers and the Editors have raised some concerns with your manuscript and we would like to invite you to revise your manuscript to address them.

As you will know, an important feature of PRSB, is to maximise accessibility and understanding across a broad readership. In that vein, I endorse the comments and suggestion from referee one, concerning the value of an overall schematic, and encourage you strongly to consider including one.

We agree an overall schematic is helpful. We therefore added a summary of the experimental design for the main experiment to Fig. S1, which previously depicted only the most complicated of our experiments. We have kept this figure in the supplement because we are already close to the length limit, and we think the revised main manuscript is sufficiently clear on these aspects that most readers will benefit more from having the data figures in the main text, supported by supplementary schematics for any readers seeking clarification or extra detail on these aspects.

We do not allow multiple rounds of revision so we urge you to make every effort to fully address all of the comments at this stage. If deemed necessary by the Associate Editor, your manuscript will be sent back to one or more of the original reviewers for assessment. If the

original reviewers are not available we may invite new reviewers. Please note that we cannot guarantee eventual acceptance of your manuscript at this stage.

Research ethics:

Use of animals and field studies:

It is a condition of publication that you make available the data and research materials supporting the results in the article (<https://royalsociety.org/journals/authors/author-guidelines/#data>). Datasets should be deposited in an appropriate publicly available repository and details of the associated accession number, link or DOI to the datasets must be included in the Data Accessibility section of the article (<https://royalsociety.org/journals/ethics-policies/data-sharing-mining/>). Reference(s) to datasets should also be included in the reference list of the article with DOIs (where available).

Please submit a copy of your revised paper within three weeks. If we do not hear from you within this time your manuscript will be rejected. If you are unable to meet this deadline please let us know as soon as possible, as we may be able to grant a short extension.

Best wishes,

Professor Gary Carvalho
mailto: proceedingsb@royalsociety.org

Associate Editor

Comments to Author:

Thank you for resubmitting your revised manuscript. This was sent to one of the original reviewers and 2 new reviewers. As you will see the original reviewer is entirely satisfied, and the 2 new reviewers raise a few minor issues that can be addressed in revision.

Reviewer(s)' Comments to Author:

Referee: 2

Comments to the Author(s).

I thank the authors for clearly addressing each of my concerns. While I agree with Referee 1 that it would be interesting to get a better view of what is driving the colonization resistance observed, I think as it stands this work already provides a useful contribution to the field and merits being published here as is. I do still think if space allows the manuscript would benefit from including the schematic in the main text, but I appreciate this is a matter of taste.

As described above, we prefer to keep the schematic in the supplement because of the length, particularly now we further clarified the text in methods and results in response to Referee 3's suggestions. We think the main text and figures are sufficiently clear that the schematic will only be important for a subset of readers seeking extra clarification or details, whereas the data figures are more central to supporting the story. If the editor really prefers, we move the schematic to the main text we are happy to hear suggestions, but our preference is to keep this in the supplement. Finally, we added a new panel to this

schematic, depicting the design of the main experiment (corresponding data in Fig. 1) for further clarity.

Referee: 3

Comments to the Author(s).

Major comments

Through-out the text you talk about “adaptation” of the community and how this affects invasion success. Reading the text my impression is what you have in mind is adjustment of the community performance by ecological means, like species sorting or ecological drift. I don't think you mean adaptive evolution in terms of heritable modifications of the phenotypes through genetic changes. As you maintained your cultures for 72h, it may be a reasonable assumption that mainly ecological forces drive community change. If this is true, your terminology through-out the text is misleading. In case you indeed think adaptive evolution occurred in your experiment, I would be more critical asking for proper measures how much fitness of particular species changed and how much your focal species evolved. While I agree that within 72h evolution may not be very dominant, I don't think it can be entirely excluded! We have good indication that bacteria can rapidly evolve even within a few generations and that this can have effects on community functions. Not looking for it doesn't mean it didn't happen! This should be at least discussed and we have recent literature exploring evolution in complex bacterial communities. I see this was a decent amount of work so I wouldn't straight ask for additional experiments. If you are sure no evolution occurred, you probably can argue out of that. How many generations you think happened during these 72h hours? When I went through the text the first time, I thought you calculated a generation number; reading it again I was unable to spot it.

We thank the reviewer for this interesting point. We agree "adapted" could be misinterpreted, and have now revised this throughout to "conditioned" (explained first on L172). Our 16S data indeed indicate species sorting was prevalent in our experiment, but the reviewer is correct that this does not rule out possible evolution within individual taxa. We now note this in the discussion (L368). We refrain from estimating a generation number for the focal strain because it is in some cases driven to extinction, and in others this is problematic because of the balance of births/deaths, discussed in the introduction (L79) and discussion (L414).

Minor comments

L48: Well, it may also depend on chance, initial frequencies (is this a single individual of the invader, or a large population) and the ability for rapid evolution!

We agree there are other factors that can be in play here, so we rephrased this sentence to be less exclusive (L45).

L64: I would assume that high diversity isn't the sole driver, but how well the resident community capitalizes resources needed by the invader, which of course is a function of biodiversity. This goes into the direction of functional redundancy in microbial communities. With high levels of redundancy, invasion should be more difficult compared to strong niche partition. This now should also interact with available resources. I would see a close interaction between the invading strain properties and the resident community...

We revised the text in the introduction (L65) to emphasize the importance of resources being shared with/required by the invader, and in the discussion we refer to the possibility of functional redundancy, implied by our finding that total diversity can drop but resistance to invasion remain (L368).

L71: However, when resources become scarce, competitive power of resident communities may also decrease to outcompete the invader (reduced cell densities). I guess there are two sides of the coin. Maybe represent these two less biased.

We agree with the principle here, which is accounted for in our results by our measurement of total bacterial densities (remained high, as they probably are in most gut microbiomes as well). Due to space we do not think this needs to be gone into in detail here, though we did clarify again that it is competition for resources shared by extant competitors and the invader that matter (L71).

L73: There are several well established examples (Goldford Science 2018, Lawrence PlosBio 2012, Fiegna ProcB 2015) that cross feeding has effects on coexistence.

We agree crossfeeding is an important type of interaction. We do not go into more detail here because the context of our questions and key findings is more in scenarios where the net effect of community-level interactions is competitive/suppressive for an invader, although we do mention crossfeeding as a potential modulator of these interactions on L72. We provide Rakoff-Nahoum 2016 Nature as an example here because it is specifically about the gut microbiota and therefore more relevant to this study.

L 134: It would be helpful to briefly describe how "Hungate" tubes work. While there is a reference this "abiotic" condition is crucial for the study question.

We have added "individually sealed anaerobic test tubes" to the methods (L134).

L 137: What was the CFU of the live slurry?

Total bacterial abundances in the live slurry are given in Figure 2, as estimated by flow cytometry. Note CFU counts obtained by plating from live slurry would almost certainly underestimate total community abundances, because many species cannot be cultivated individually (great plate count anomaly). Abundance of the focal strain in live slurry was estimated by plating, and this is given in Figure 1.

L145-148: There is an extrapolation from total cell density measured by flow cytometry towards ratio of the invader measured by plating. As I understand it, the community density was not determined by plating. I can see that double plating would be too much work. Are 100% of the invader cells surviving the plating process? In the ideal case there should be a 1:1 line, cell counts from the flow and the CFU, which may be a point in the Supplements.

As above, it would not be very informative to plate entire communities. We agree it could help interpret data on focal strain abundances to know how they relate to abundances of other species estimated by flow cytometry. Our data do not permit a direct comparison here, although this does not affect any of our key findings (which are about variation of one measure or the other across treatment groups). In the one part of the manuscript where we consider both types of information, we do so very cautiously (Table S1 and associated text). Furthermore, both methods are extremely widely used and when they have been tested alongside each other in similar set-ups, other authors have found strong agreement (e.g., Roussel et al 2018 App. Micro. Biotech. doi.org/10.1007/s00253-018-9380-z).

L170: Not sure if the word "adapted" is appropriate. I guess the authors don't use it in an evolutionary context but mean the communities have ecologically stabilized? But see comments on evolution....

We agree that "adapted" could potentially be misinterpreted. We have now rephrased throughout to "conditioned to the microcosm environment" or abbreviated versions of this phrase, which we explain on L172.

L173: I didn't really understand how the swapping was done. Supp Fig. S1 is not very informative on this. Maybe redraw the figure or give a bit more info in the text. There is more information in the Supplementary Methods but still it doesn't become clear to me.

The wording "fresh community+fresh" and "adapted community+spend" is also confusing. Is there also fresh community+spend? When you talk about "adapted community", that has

been cultured for 72h right? "Fresh community" would then be grown for 24 h? Well, maybe this becomes clearer in the Results...

We have now adjusted the wording (avoiding "adapted") and refreshed the explanation in the methods (L170-191) and the legend of Figure S1.

L178: When you say "after 72h" you mean the 72h grown cultures, right?

Yes, we now clarify this in the text (L180).

L199: I guess a specific region of the 16S gene was sequenced? Ah given in the Supplements... Why not just adding it in the Methods?

Some details that are not essential for most readers are given in the supplement because of space limitations.

L220: By the way fresh medium is unused medium, right? So, you added the bacterial community and directly removed it?

Fresh medium is the basal medium added to faecal slurry; which either contains bacteria or is sterilized (with/without community = live/sterilized slurry). We hope this is clearer now in the context of the new panel of Fig. S1 and the revised methods.

L211: What is the difference between "Donor and Community" in the fixed effects of your model. The community was derived from the donors, right? I guess you have some nasty correlation situation where you track Community over Time but Community is correlated with Donor... While "Replicate" and "ID" will address the temporal correlation, I guess you need either additional random terms telling the model community is nested in Donor or correlation structures to address the non-independency between Community and Donor. Maybe nlme is better equipped to include the correlation structure.

We did not explain the different treatment groups and factor levels clearly enough before. "Community" has two levels: with and without (live and sterilized faecal slurry, prepared for each donor using faecal sample from that donor). "Donor" indicates the human donor from which each sample comes. We added a new panel to Figure S1 to clarify this, and in the methods (L214) and legend of Figure 1 (L619) we clarified the wording.

L236: The lmer will give you a slope over time! When you say the effect was strongest towards the end, where does this come from? Did you include non-linear trends indicating a change over time? Or have you used "time" as a factor? In this case you would get differences for time points compared to the starting time point.

We fitted time as a categorical factor, partly because changes over time are in some cases non-linear. In the revised methods we now clarify that time was a fixed factor in this model.

L243: According with my comment above, can it be excluded that donor1 and community1 are independent from each other? With donor1 you refer to slurry from donor1, right?

As above, we hope the clarifications to Figure S1 and the text make it easier to understand the experimental design: the community treatment indicates live/sterilized slurry for each donor, rather than indicating communities 1, 2 and 3.

L249: Would this mean there was active killing by the community? If the focal strain density wouldn't increase but the community would grow I would think competition constrains the invader but there is no active killing. Of course, here we run into the problem that the community may simply grow at a grow/death balance, which is likely hard to exclude...

We can infer that deaths outnumber births for the focal strain, but observing direct killing is more challenging. We discuss the births/deaths issue in the introduction (L79) and discussion (L414), and also the plausibility of direct killing indicated by our 16S data showing resident enterobacteriaceae were abundant (L268) and past evidence of such populations carrying Type VI secretion systems and similar anticompertitor weaponry (L407).

L263: A bit of poor luck that you invader was hitting the ecological dominant species in your experiments. Now this interacts with my comment above. Resident Ecoli are increasing while invading Ecoli die, is there active killing or sever competition?

Given the dependence on local nutrient conditions, a role for resource competition seems likely, though in the revised discussion we do not rule out an additional, non-exclusive possibility of direct killing (L405-419).

L278: This also indicates non-independency between community and donor...

As above, we hope the revised explanation of experimental design and model fitting (L214) have cleared this up. The model fitted here accounts for both which human donor the sample used in each microcosm came from (donor), and whether it was live or sterilized (community).

L306: Guess you mean compared to control rather than sterile slurry.

We hope the difference between control (basal medium) and sterilized slurry (basal medium plus sterile faecal slurry) treatments is now clearer. Here we are referring to sterilized slurry treatments, which is the best control for the community (live slurry) treatments because it is prepared from the same faecal sample, but is sterilized.

L315: In line 310 you say "environment became NOT nutrient poor". This is a bit confusing. It became poorer but not so much it would prevent growth. Maybe re-word a bit.

We have rephrased here (L314).

L326: and you grew them for XX days? I mean your fresh community would become adapted after 72 hours...

The fresh community has not been grown for 72 hours, as clarified in the revised schematic.

L337: I am not sure you really "showed" that colonization success depends on interaction. I feel you have good indication but can also see alternative explanations. Neither it is clear "demonstration" L339...

This text describes only the result, rather than the explanation. We agree it is important not to overstate the mechanistic implications, but we think the statistical result given in this sentence is supported by the data.

L343: Have you really convincing indication that nutrients get depleted? It could be more an exhaustion of easily accessible resources with still much carbon for growth available bound in recalcitrant form.

The relatively nutrient poor state in these microcosms is indicated strongly by the supernatant experiments (Figure 4).

L361: "Less diverse communities seem to be harder to invade". This connects with my main criticism of this work. When members of the community evolved, in terms of fitness change and resource use, more complete pre-emption of niches would constrain invaders even when non-functional diversity is lost. Are your communities going though adaptive evolution or is this some "community adaptation process" by means of species sorting and ecological drift?

Thank you for this interesting point. Our 16S data are consistent with species sorting/changes in the relative abundances of different taxa, which we now discuss in the text on L368-372. Additionally, we do not rule out the possibility of evolution within individual taxa during the experiment (L372), although this would not alter our main conclusions.

L369: What else would we need? Just remind the reader!

We added a little more detail here (L378).

L382: You should compare your findings to previous results from the human microbiome consortium. There, host diet and physiological status were also discussed. We added a reference to the Human Microbiome Project in the discussion.

Figures:

Fig. 1 These are means of 3 replicates, right? Maybe add error bars!

Each line depicts the time series for an independent biological replicate of the experiment.

So, this means you added communities 1-3 to sterile slurry from donor 1 and similarly to slurries from d2-d3? This also only became clear to me after looking into the figure...

No, for each donor there are two treatments: with and without community, both prepared using the same stool sample (live and sterilized versions of the same faecal slurry, separately prepared for each donor). We hope the revised Figure S1 and clarifications in the text have cleared this up.

This now goes back into my comment on stats what variables are donor & community...

What I am somehow missing is how communities did in "Control" medium. You didn't do this, right?

No, control medium is sterile basal medium, without resident microbiota.

Fig2: It is good that communities don't die and interesting to see they don't grow biomass anymore. Anything else we learn from the figure? If that figure would go into the supplements I wouldn't miss it...

We agree that stability of communities at high density is important, and think this data is also important for interpreting the dynamics for the focal strain. Therefore, we keep it in the main text.

Fig. 3: "Adapted 1-3" means three replicates from last time point, right? Yes.

Or is this pooled reps for 3 time points? No.

Fig. 4: This is final density after 24hours? No. This is the difference between CFU after 24h of growth and CFU at the start of the experiment.

Correct me, but in the control medium nutrients were not pre-used (unless the invader was grown in it) but the slurry is metabolized already when you got it from the donor.

Control medium is simply sterile basal medium, whereas community and community-free treatments for each donor are live/sterile versions of faecal slurry prepared using the sample from that individual donor (by suspending the sample in peptone wash and then basal medium, as described in the methods on L123-L131). We hope the new panel in Fig. S1 helps clarify the overall experimental design.

I guess control supernatant from communities would be helpful... In general, we learn from this that the slurry still has lots of nutrients inside and the additional 72 hours don't change this much.

Have 72 hours been enough to run the experiment when growth is obviously slow?

While it would of course be interesting to see what happens over weeks or longer, three days was clearly enough to observe an increasing suppression of the focal strain to densities >1000 times lower than at the start (but crucially, only in the community treatments), and in some cases to extinction.

Referee: 4

Comments to the Author(s).

In their manuscript "Microbial community composition interacts with local abiotic conditions

to drive colonization resistance in human gut microbiome samples” Baumgartner et al. elucidate how biotic and abiotic factors shape the resilience of human gut microbial communities against invasion through a focal *E. coli* strain. It hence provides novel insights into an important question associated to human health.

While a clear mechanistic understanding into how exactly resilience of microbial communities is shaped is lacking and outside the scope of this study, the manuscript provides a great basis to explore the hypothesized mechanisms in depth in follow up studies.

The manuscript is well written, easy to understand and logically structured. The authors rely on a set of simple, but very well designed experimental assays that built upon each other to increase the complexity of tested biotic and abiotic variables.

The authors have replied well and sufficiently to the comments raised by the reviewers and consequently improved the quality of the manuscript. I believe that there are only a few minor remaining points to address.

L. 23: Rephrase: Community composition was only measured after 0 and 72h. As dynamics in composition shifts were not monitored across this period, it is not possible to associate shifts in community composition to later time points exclusively.

We have rephrased this as suggested (L23).

L. 326-328: rephrase, to ensure the clear statement of this important result
Rephrased (L332-333)

Fig. 3: Ensure capitalization of “Human Donor” and “(Community, Replicate)” is consistent between panels.

Revised as suggested.